# Motor cortex activity across movement speeds is predicted by network-level strategies for generating muscle activity

**Shreya Saxena**[1,2,3,4,5†], **Abigail A Russo**[2,6†], **John Cunningham**[2,3,4,5], **Mark M Churchland**[2,3,6,7]*

[1]Department of Electrical and Computer Engineering, University of Florida, Gainesville, United States; [2]Zuckerman Mind Brain Behavior Institute, Columbia University, New York, United States; [3]Grossman Center for the Statistics of Mind, Columbia University, New York, United States; [4]Center for Theoretical Neuroscience, Columbia University, New York, United States; [5]Department of Statistics, Columbia University, New York, United States; [6]Department of Neuroscience, Columbia University, New York, United States; [7]Kavli Institute for Brain Science, Columbia University, New York, United States

**Abstract** Learned movements can be skillfully performed at different paces. What neural strategies produce this flexibility? Can they be predicted and understood by network modeling? We trained monkeys to perform a cycling task at different speeds, and trained artificial recurrent networks to generate the empirical muscle-activity patterns. Network solutions reflected the principle that smooth well-behaved dynamics require low trajectory tangling. Network solutions had a consistent form, which yielded quantitative and qualitative predictions. To evaluate predictions, we analyzed motor cortex activity recorded during the same task. Responses supported the hypothesis that the dominant neural signals reflect not muscle activity, but network-level strategies for generating muscle activity. Single-neuron responses were better accounted for by network activity than by muscle activity. Similarly, neural population trajectories shared their organization not with muscle trajectories, but with network solutions. Thus, cortical activity could be understood based on the need to generate muscle activity via dynamics that allow smooth, robust control over movement speed.

**\*For correspondence:** mc3502@columbia.edu

†These authors contributed equally to this work

**Competing interest:** The authors declare that no competing interests exist.

## Editor's evaluation

This elegant study furthers our understanding about the mechanisms by which distributed systems control rhythmic movements of different speeds. The authors trained an artificial recurrent neural network to produce muscle activity patterns similar to those that monkeys generate when performing an arm cycling task at different speeds. The dominant patterns in the neural network do not directly reflect muscle activity, and these dominant patterns do a better job than muscle activity at capturing key features of neural activity recorded from the monkey motor cortex in the same task. In addition to the main result, the study provides a particularly clear example of how thinking in terms of network dynamics can naturally explain empirical observations in terms of the computation being performed.

## Introduction

We can often perform the same action at different speeds. This flexibility requires multiple adjustments: scaling the pace of muscle activity while also altering its magnitude and temporal pattern. Given that movement speed can typically be adjusted continuously, it seems unlikely that the brain

employs a separate solution for each speed. What might a unified solution look like, and how might it be reflected in the structure of neural population activity?

How this question should be approached interacts with a more basic issue: our understanding of the relationship between cortical activity and movement. This understanding has recently been in flux. Hypotheses that regard neural responses primarily as representations of controlled variables (for example, hand velocity or muscle activity) have been challenged by hypotheses that view neural responses as reflecting the evolution of movement-generating dynamical systems (*Bruno et al., 2017*; *Pandarinath et al., 2018*; *Sussillo et al., 2015*; *Russo et al., 2018*; *Michaels et al., 2019*; *Churchland et al., 2012*; *Lillicrap and Scott, 2013*; *Shenoy et al., 2013*; *Hall et al., 2014*). This 'network-dynamics' perspective seeks to explain neural activity in terms of the underlying computational mechanisms that generate outgoing commands. Based on observations in simulated networks, it is hypothesized that the dominant aspects of neural activity are shaped largely by the needs of the computation, with representational signals (for example, outgoing commands) typically being small enough that few neurons show activity that mirrors network outputs. The network-dynamics perspective explains multiple response features that are difficult to account for from a purely representational perspective (*Churchland et al., 2012*; *Sussillo et al., 2015*; *Russo et al., 2018*; *Michaels et al., 2016*). Yet the purely representational perspective did convey a practical advantage: it readily supplied predictions in new situations. For example, when movement speed increases, a neuron that represents velocity should simply reflect the increased velocity. A neuron that represents muscle activity should simply reflect the new pattern of muscle activity. Can the dynamical perspective make similarly clear (and presumably different) predictions?

Computations performed by artificial – and presumably real – networks can often be described by flow-fields in state space. Because these flow-fields shape population trajectories, and because empirical population trajectories are readily plotted and analyzed in state-space, this affords a means of comparing data with predictions. Goal-driven networks – that is, networks trained to perform a computation intended to be analogous to that performed by a biological neural population (*Zipser and Andersen, 1988*; *Yamins et al., 2014*; *Lindsay and Miller, 2018*) – are increasingly used to model computations requiring internal or external feedback (*Mante et al., 2013*; *Sussillo et al., 2015*; *Maheswaranathan et al., 2019*; *Sohn et al., 2019*; *Kao et al., 2020*; *Michaels et al., 2019*; *Rajan et al., 2016*; *Perich and Rajan, 2020*). Although such models typically lack detailed anatomical realism, they yield solutions that can be understood through reverse engineering and afford comparisons with data. The continuous control of movement speed presents a situation where this approach might hope to generate particularly clear predictions because the range of possible solutions is likely to be constrained: network dynamics must be 'well-behaved' in some key ways. First, the network solution should change continuously with speed. The alternative – a distinct solution for each speed – is likely incompatible with the fact that speed can be continuously adjusted. Similarly, network solutions should gracefully interpolate between speeds used during training. Finally, noise should not cause large errors or the generation of the wrong speed. All these forms of robustness are expected to benefit from smoothly varying underlying dynamics, resulting in population activity with very low 'trajectory tangling' such that similar states are never associated with dissimilar derivatives (*Russo et al., 2018*).

We sought to determine whether these constraints allow network solutions to predict empirical responses. We trained monkeys to perform a cycling task at different speeds, and trained recurrent neural networks to generate the observed muscle-activity patterns, with speed instructed by a graded input. All such networks adopted a strategy with two prominent characteristics. First, the network trajectory for every speed resembled an ellipse in the dominant two dimensions, and was strikingly invariant to the shape of the muscle-trajectory output. Second, individual-speed trajectories were separated by a translation in a third dimension. We confirmed that this structure reflected stable input-dependent limit cycles, maintained low trajectory tangling, and reflected the need for solution continuity. We then compared network solutions with population activity recorded from motor cortex. At both single-neuron and population levels, neural activity matched network activity more closely than it did muscle activity. Neural trajectories had the same within-speed and across-speed organization as network trajectories, and also displayed very low trajectory tangling. Thus, while the dominant structure of motor cortex activity does not resemble that of muscle activity, it closely matches solutions adopted by networks that employ dynamics to generate muscle activity. These findings demonstrate

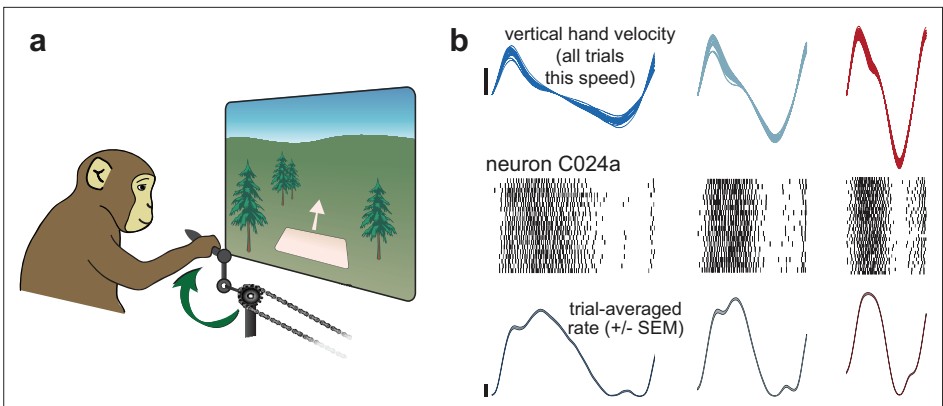

**Figure 1.** Illustration of behavior and neural activity. (**a**) Schematic of the task, which required cycling at a range of different speeds for juice reward. (**b**) Behavior and neural activity for one example neuron (C024a) and three cycling speeds. Data are for Monkey C, cycling forward. All data are plotted after temporal alignment within each speed bin. For visualization, plots show just over one cycle. Top: vertical hand velocity for every trial in the three speed bins. These overlap heavily, forming an envelope spanning trial-to-trial variability. Rasters show spikes for ~25 trials per speed bin. Color denotes speed bin, with faster speeds in *red* and slower in *blue*. Bottom: *colored* traces plot the trial-averaged firing rate. Shaded envelopes for firing rates indicate the standard error of the mean (SEM). Vertical calibration indicates 10 spikes/s.

that features of good network-level solutions can provide strong and successful predictions regarding empirical neural activity.

## Results
### Task, behavior, muscle activity, and single neuron responses
We trained two rhesus macaque monkeys to grasp a pedal with their hand and cycle for juice reward (***Russo et al., 2018***; ***Russo et al., 2020***). Cycling produced movement through a virtual landscape (***Figure 1a***). Landscape color indicated whether forward virtual movement required 'forward' cycling or 'backward' cycling, performed in alternating blocks. Monkeys cycled continuously to track a moving target, in contrast to ***Russo et al., 2018*** and ***Russo et al., 2020*** where targets were stationary. Target speed was constant within long (30 second) trials. There was natural within-trial variation in cycling speed. Monkeys would often slow down modestly for a few cycles, fall behind the target, then speed up modestly for a few cycles. As a result, there was an essentially continuous range of cycling speeds within each session. To allow analysis, data were divided into individual cycles, beginning and ending with the pedal oriented straight down. Cycles were classified into eight speed bins, spaced in ~0.2 Hz intervals. Bins were chosen to span the relevant range for each direction and monkey and are thus labeled by number (1–8, from ~0.8–2.1 Hz) rather than frequency. Cycles within a bin were scaled to have the same duration, but the values of key variables (e.g., velocity, firing rate) were computed before scaling and never altered. Cycles where there was a large change in speed were not analyzed, ensuring that trials within a bin had nearly identical velocity profiles (***Figure 1b***).

Well-isolated single units (49 and 52 neurons for Monkeys C and D) were sequentially recorded from motor cortex, including sulcal and surface primary motor cortex and the immediately adjacent aspect of dorsal premotor cortex. Cycling evoked particularly strong responses. Nearly all neurons that could be isolated were task-modulated. A clear, repeatable structure in spiking activity was apparent across single cycles (***Figure 1b***, raster plots). We computed the firing rate of each neuron by averaging across cycles within a speed bin (median of 84 and 80 trials, monkey C and D). This yielded a time-varying trial-averaged firing rate (***Figure 1b***, bottom) and standard error (envelopes are narrow and thus barely visible).

Neural responses were heterogeneous (***Figure 2***). In some cases, response magnitude was largest at the fastest cycling speeds (***Figure 2a***, top). In others it was largest for the slowest speeds (***Figure 2c***, bottom) or changed only modestly (***Figure 2a***, bottom). A given neuron could show different relationships with speed depending on cycling direction (***Figure 2b***). Response complexity was not due to

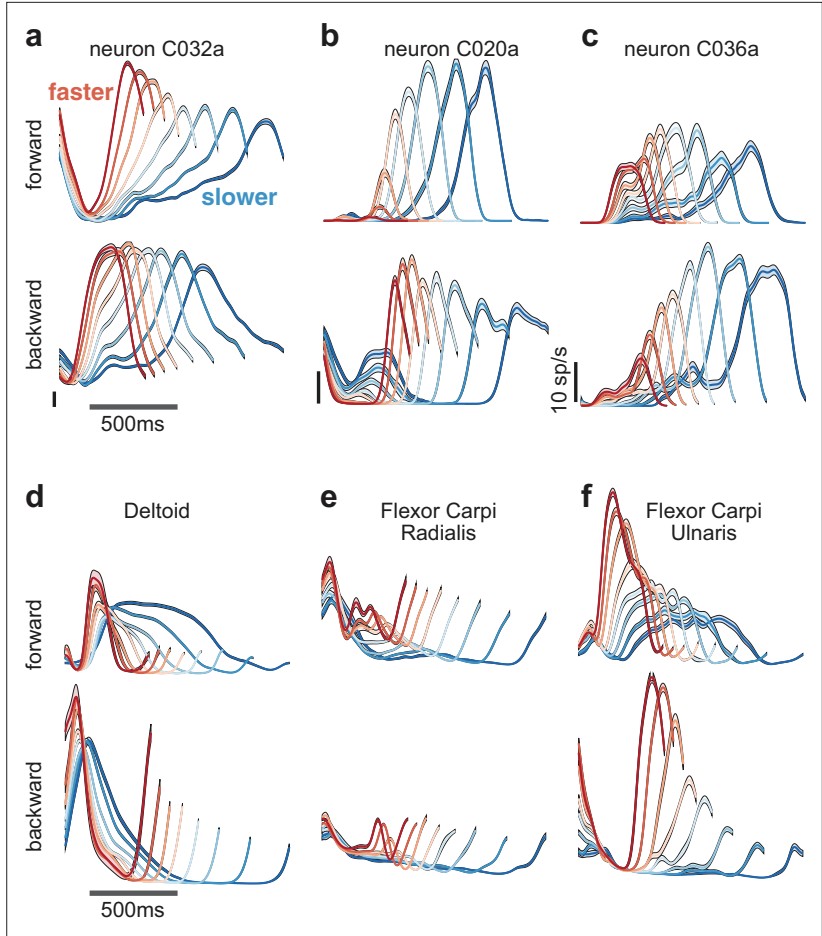

**Figure 2.** Speed-dependent neural and muscle activity patterns during cycling. (**a–c**) Neural and (**d–f**) muscle activity across all eight speeds. Mean across-trial firing rate (or mean across-trial muscle activity) is plotted versus time for each speed bin. Envelopes give SEM across trials. Each panel plots activity for one example neuron or muscle. In each panel, data for forward cycling is plotted on top and data for backward cycling is plotted on bottom. All data is for monkey C. Traces are colored red to blue according to speed. All vertical calibrations for neural activity are 10 spikes/s. Muscle activity scale is arbitrary but preserved across all traces for a given muscle.

noise; standard errors were typically small (flanking envelopes are narrow enough to often be barely visible). Across all neurons, 33 showed an increase in response magnitude with speed for both cycling directions, 24 showed a decrease for both, and 44 scaled oppositely. Responses could be monophasic or multiphasic depending on the neuron, cycling direction, and speed. Temporal response patterns were typically similar, other than temporal scaling, for neighboring speed bins. More dramatic changes in temporal response pattern could occur across the full range of speeds. For example, responses could be multiphasic at slow speeds but monophasic at faster speeds (*Figure 2b*, *bottom*). Responses of major upper-arm muscles were also heterogeneous (15 and 33 recordings in monkeys C and D; *Figure 2d–f*). A key question is whether neural activity reflects structure beyond that present in the muscles and, if so, whether that structure can be predicted by network-level solutions.

## Generating predictions using goal-driven modeling

We trained recurrent neural networks, via backpropagation through time, to produce the empirical patterns of muscle activity across speeds (*Figure 3a*). Target outputs were the projection of muscle population activity onto its top 6 principal components (PCs). Speed was instructed by the magnitude of a simple static input. This choice was made both for simplicity and by rough analogy to the loco-motor system; spinal pattern generation can be modulated by constant inputs from supraspinal areas

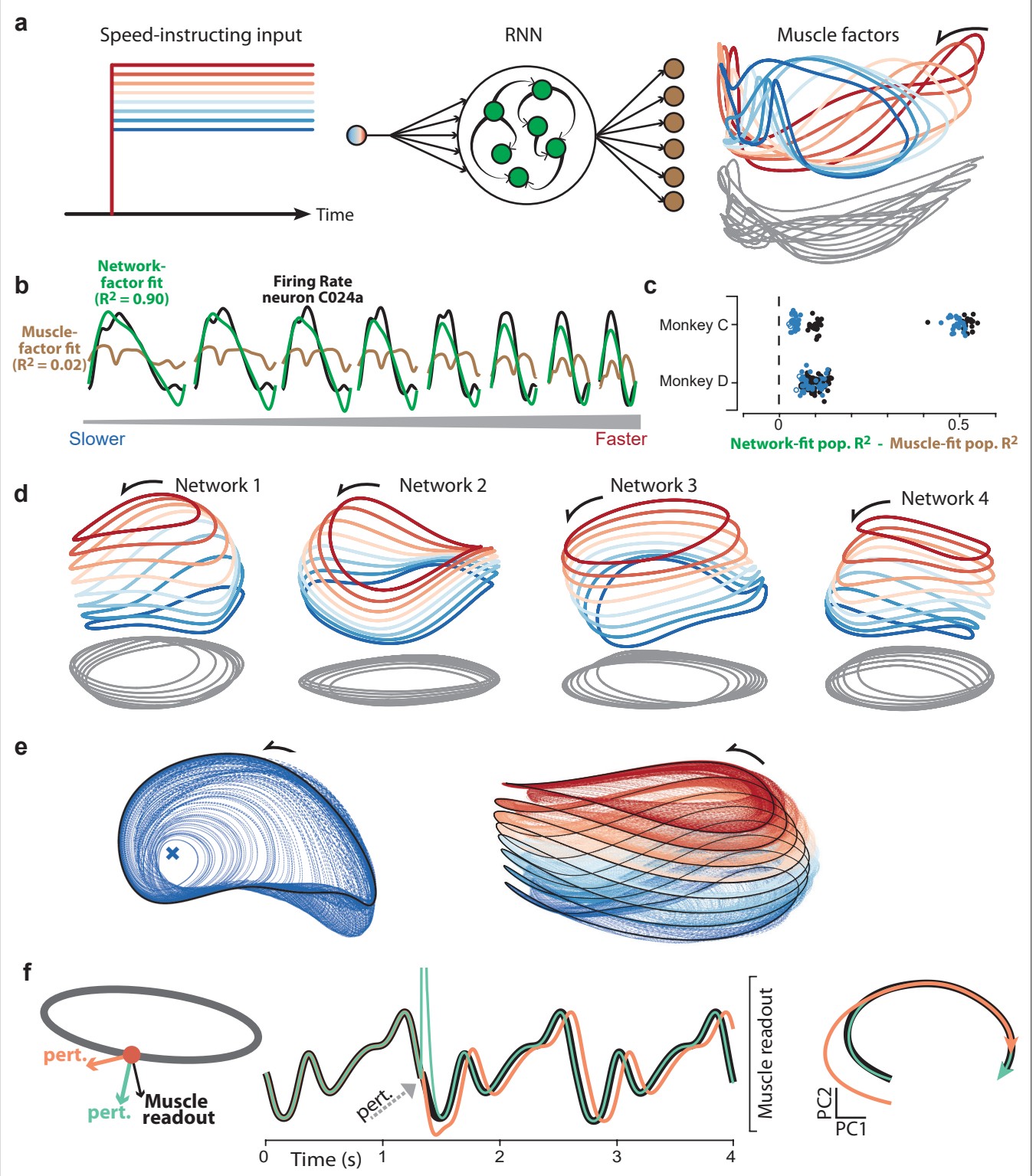

**Figure 3.** Goal-driven neural network solutions. (**a**) Recurrent neural networks (RNNs) received a constant input whose level instructed cycling speed. Network-output targets were the top six muscle population factors (the projection of muscle population activity onto each of the top six PCs). Trajectories for the top three factors are plotted in state space, colored blue-to-red with increasing speed. Network outputs (not plotted for simplicity) were nearly identical to the target factors. (**b**) Example, for one neuron and all speeds, of fitting single-neuron activity with network factors (*green*) or muscle factors (*brown*). Fits were obtained by regressing the neuron's activity against the top three factors (network or muscle). The $R^2$ values shown here correspond to the example neuron. For the analyses below, we computed the population $R^2$ across all neurons. The population $R^2$ is the mean $R^2$ across

*Figure 3 continued on next page*

*Figure 3 continued*

neurons, but taking into account response magnitudes (for example, a poor fit for a low-rate neuron has less impact). (**c**) Difference in population $R^2$ for every network. The two cycling directions and two monkeys provided four network-output targets. For each we trained many networks. For each such network, the difference in population $R^2$ for network-factor fits versus muscle-factor fits is plotted as a single point. The analysis was performed twice: once when fitting using two factors (*black circles*) and once when fitting using three factors (*blue circles*). Filled circles denote $p<0.05$ (paired one-tailed t-test with n=8 speeds) when testing whether network factors provided significantly better decoding than muscle factors. (**d**) Network trajectories for four example networks projected onto global PCs 1, 2, and 3 (the dominant dimensions across all speeds). Networks employ non-identical solutions due to different weight initializations before training. (**e**) Network trajectories are stable. *Left:* The network was perturbed so that it started at different initial states. All trajectories converge to the same limit cycle. Network input corresponded to the slow speed. *Right:* Same, but repeated for all eight input levels. Trajectories converge to an input-dependent limit cycle. (**f**) Using perturbations to explore network strategy. We delivered two perturbations to the network state. The rhythm-generation perturbation (*orange*) was designed to strongly impact activity in the plane containing the elliptical path (the plane defined by the top two PCs for the speed being examined) while remaining orthogonal to the muscle readout. The muscle-generation perturbation (*green*) was designed to do the opposite: strongly impact activity in one of the muscle readout dimensions while remaining orthogonal to the top two PCs. Traces versus time (*middle*) show network output with no perturbation (*black*), and the two perturbations (*orange* and *green*). State-space trajectories (*right*) plot network population activity in the top two PCs, which capture the dominant elliptical trajectory. Trajectories begin at the time of the perturbation. Consistent with its design, the muscle-generation perturbation had no immediate impact in this plane: the *green* trajectory begins in the same location as the black (unperturbed) trajectory. This reflects the fact that the muscle-generation perturbation had a large impact in other dimensions, but almost no influence on the future evolution of the elliptical trajectory. In contrast, the rhythm-generation perturbation had a large immediate impact on the elliptical trajectory, and permanently altered its phase: the *orange* trajectory ends at an earlier phase than the *black* trajectory, even though both are plotted for the same time-span.

The online version of this article includes the following figure supplement(s) for figure 3:

**Figure supplement 1.** Trajectory tangling values for RNN trajectories vs EMG trajectories in the global (12 dimensional) subspace.

**Figure supplement 2.** Network trajectories for networks receiving input commands other than the graded-speed command used for the networks in *Figure 3*.

---

(*Grillner, 1997*). Of course, cycling is very unlike locomotion and little is known regarding the source or nature of the commanding inputs. We thus explore other possible input choices below.

Each network was trained to produce muscle activity for one cycling direction. Networks could readily be trained to produce muscle activity for both cycling directions by providing separate forward- and backward-commanding inputs (each structured as in *Figure 3a*). This simply yielded separate solutions for forward and backward, each similar to that seen when training only that direction. For simplicity, and because all analyses of data involve within-direction comparisons, we only consider networks trained to produce muscle activity for one direction at a time. Networks were trained across many simulated 'trials', each with an unpredictable number of cycles. This discouraged non-periodic solutions, which optimization might use if the number of cycles were fixed and small. We trained 25 networks for each monkey and direction (100 networks total). Of these, 96 successfully performed the task after 700,000 training iterations (24 and 25 for Monkey C, forward and backward; 22 and 25 for Monkey D, forward and backward). Success was defined as <0.01 normalized mean-squared error between outputs and targets (i.e. an $R^2 >0.99$). Because 6 PCs captured ~95% of the total variance in the muscle population (94.6% and 94.8% for monkey C and D), linear readouts of network activity yielded the activity of all recorded muscles with high fidelity. Networks differed in their initial random weights, and thus showed non-identical final solutions.

No attempt was made at detailed realism: networks employed a single cell type, all-to-all connections, and no differentiation between feedback due to local versus long-range or re-afferent sources. This allowed us to focus on the form of network-level solutions rather than their implementation. A natural first question regards the heterogenous single-neuron responses: does the network-based hypothesis provide improved explanatory power relative to the baseline hypothesis that neural activity is an upstream reflection of muscle activity? Regardless of whether this 'muscle-encoding' hypothesis is taken literally, it provides a stringent bar by which to judge model success, because neural and muscle activity naturally share many features.

Under both the network-based and muscle-encoding hypotheses, single-neuron responses are predicted to reflect a set of underlying factors: $r_i(t) = \sum_j w_{ij} x_j(t)$, where $r_i(t)$ is the time-varying response of the $i^{th}$ neuron, $w_{ij}$ are weights, and $x_j(t)$ are the factors. The two hypotheses can be compared by regressing empirical neural activity versus the relevant underlying factors: network-based factors or muscle-based factors. To identify those factors, we applied Principal Component Analysis (PCA) to the relevant population (muscle or network) and considered the top two or three

PCs. Within a cycling direction, three PCs explain >80% of the variance in both muscles and networks. Projections onto the muscle-derived PCs capture the dominant factors within the muscle population (the signals most strongly shared amongst muscles). Similarly, projections onto the network-derived PCs capture the dominant factors within the simulated network. The coevolution of the top three muscle factors is shown in state-space in *Figure 3a* (*right*). Note that while muscle factors are target network outputs, they differ from the top network factors (*Figure 3d*). It is thus possible for a neuron's response to be fit poorly by muscle factors but well by network factors (*Figure 3b*).

Neural responses were overall better accounted for by network factors (*Figure 3c*). This improvement, relative to the muscle factors, was statistically significant for the vast majority of individual networks (p<0.05 for 90/96 networks, paired t-test with n=8 speeds, comparison using three PCs). This effect was clearest when considering the dominant signals (shown in *Figure 3c* for the first two PCs - *black circles* - and first three PCs - *blue circles*) but remained present when using more PCs. For example, when using six PCs, all networks provided an improvement over the muscles for monkey C, and all but one network provided an improvement for monkey D. These advantages were maintained if we assessed generalization $R^2$ (for left out conditions) to guard against overfitting.

Although network factors essentially always provided a better basis for explaining neural activity, the degree to which this was true varied across monkeys, cycling directions, and networks. For monkey C, the improvement in $R^2$ ranged from 0.03 to 0.12 for backward cycling and from 0.41 to 0.55 for forward cycling. For monkey D, $R^2$ differences ranged from 0.05 to 0.14 for both cycling directions. This range highlights that different networks found quantitatively different solutions. The same is presumably true of the biological networks in the two monkeys. The critical question is thus not whether factors match exactly, but whether the computationally important features of network solutions are shared with the neural population responses. To address this, we seek to understand what network solutions look like and why they are successful.

## Understanding network solutions

Recurrent-network solutions tend to have the following useful characteristic: a basic understanding does not require considering every unit and connection, but can be obtained by considering a smaller number of factors, each a weighted sum of the activity of all units (*Sussillo and Barak, 2013*; *DePasquale et al., 2016*; *Maheswaranathan et al., 2019*; *Mante et al., 2013*). By ascertaining how and why population trajectories behave in this 'factor space', one can often determine how the network solves the task. Assuming factors are defined wisely, the response of each individual unit is approximately a weighted sum of factors. Thus, if one understands the factors, individual-unit responses are no longer mysterious. There are many reasonable ways of obtaining factors, but PCA is commonly used because it ensures factors will be explanatory of single-unit responses ('maximizing captured variance' is equivalent to 'minimizing single-neuron reconstruction error'). We used PCA above to identify network factors and show that they are explanatory not only of their own single-unit responses, but of empirical single-neuron responses as well. However, knowing that network factors are quantitatively explanatory means little unless one also understands why those factors behave as they do. Thus, we now turn to the task of understanding network solutions in the factor-space obtained by PCA.

The major structural features shared by all solutions were visible in the top three PCs (*Figure 3d*). The network trajectory for each speed always resembled an ellipse (revealed by the gray 'shadows'). An elliptical trajectory is not inevitable; the rhythmic task ensures a repeating trajectory but not an ellipse. Indeed, muscle-factor trajectories (*Figure 3a*, *right*) were not particularly elliptical. For every network, trajectories were better fit by ellipses than were the corresponding muscle trajectories (p<0.05; paired, one-sided Wilcoxon signed rank test comparing $R^2$ values). Elliptical network trajectories formed stable limit cycles with a period matching that of the muscle activity at each speed. We confirmed stability by altering network activity, something only rarely possible in physiological circuits (*Bruno et al., 2017*). When the network state was moved off a cycle, the trajectory converged back to that cycle. For example, in *Figure 3e* (*left*) perturbations never caused the trajectory to permanently depart in some new direction or fall into some other limit cycle; each perturbed trajectory (*blue*) returned to the stable limit cycle (*black*).

The ability of elliptical network trajectories to generate non-elliptical muscle trajectories seems counter-intuitive but is expected (*Russo et al., 2018*). Elliptical trajectories set the basic rhythm and

provide the fundamental frequency, while additional 'muscle-encoding' signals that support complex outputs occupy dimensions beyond just the first three PCs. To illustrate, we consider a single cycling speed (*Figure 3f*) and perturb network activity in one of two directions. 'Rhythm-generation perturbations' (*orange*) overlapped with the plane of the elliptical trajectory but were orthogonal to the muscle readout. 'Muscle-generation perturbations' (*green*) overlapped with a muscle readout direction but were orthogonal to the elliptical trajectory (the perturbed direction involves PCs beyond the first three and is not the same direction that separates trajectories across speeds).

As expected, muscle-generation perturbations caused a large immediate change in muscle readout (middle subpanel, *green*). Readouts then rapidly returned to normal and the rhythm continued at its original phase. In the first two PCs, the elliptical neural trajectory was nearly unchanged following the perturbation (*green* and *black* trajectories overlap almost perfectly). Thus, muscle-generation perturbations impact a direction in network state-space that is critical for network outputs but interacts little with the dynamics that set the overall rhythm.

Rhythm-generation perturbations had only a small immediate effect on readouts, yet permanently altered the phase of network activity. This was reflected in both the readout (middle subpanel, *orange*) and the elliptical trajectory (right subpanel). Thus, the network solution involves two components. The phase of the elliptical trajectory sets the phase of the output. Muscle readouts draw from the elliptical trajectory, but also draw heavily from orthogonal dimensions that contain smaller higher frequency signals. These smaller off-ellipse features allow the network to generate non-sinusoidal activity patterns that differ across muscles. This is a natural strategy that allows a simple, stable, elliptical trajectory to generate multiple temporally structured outputs (*Russo et al., 2018*; *Lindén et al., 2021*). Trajectories are stable only when the speed-instructing input is present; trajectories return to baseline (along with readouts) when that input is extinguished (not shown).

For all networks, a translation separated limit cycles for different speeds. Network input determined which of these 'stacked' limit cycles was stable (*Figure 3e*, *right*). Stacking of elliptical trajectories is a natural strategy for adjusting speed (*Maheswaranathan et al., 2019*; *Sussillo and Barak, 2013*) for two reasons. First, unlike an arbitrary dynamical system, a recurrent network has no straightforward mechanism allowing an additive input to scale flow-field magnitude. Altering trajectory speed thus requires moving to a different region of state-space (*Remington et al., 2018*). Second, the network's target output (muscle activity) does not simply speed up for faster speeds; it changes magnitude and temporal pattern. It would thus be insufficient to traverse the same trajectory more rapidly, even if this were possible. By shifting the overall elliptical trajectory, the network can change both output frequency and output pattern.

The stacked-elliptical network solution can also be understood at a more abstract level: it provides low trajectory tangling. High trajectory tangling is defined as different moments sharing a similar location in state-space but with very different derivatives. In networks where the present state determines the future state via internal recurrence or external feedback (i.e. in a network that relies on strong dynamics) smooth noise-robust solutions require avoiding trajectory tangling (*Russo et al., 2018*). Our networks had to rely on strong dynamics because their input lacked sufficient temporal structure to specify the output. Accordingly, all networks adopted solutions where network trajectory tangling was dramatically lower than muscle trajectory tangling (paired, one-tailed t-test; $p<10^{-10}$ for every network; one example network shown in *Figure 3—figure supplement 1*). Elliptical limit cycles ensured low trajectory tangling within a given speed (a circle is the least-tangled rhythmic trajectory) and separation between limit cycles ensured low tangling between speeds. This latter fact can be appreciated by considering the implications of traversing the same trajectory at different speeds: the same set of states would be associated with different derivatives.

A natural solution for maintaining low tangling is thus elliptical limit cycles separated by a translation. We saw this solution in all networks described above and also when networks were trained in the presence of noise and thus forced to adopt noise-robust solutions. The solution not only conveys noise-robustness, it also allows continuous control of speed. Inputs between the trained input levels produced trajectories 'between' those shown in *Figure 3d*, which in turn generated outputs at intermediate frequencies. Similarly, a ramping input produced trajectories that steadily shifted and increased in speed (not shown). Thus, networks could adjust their speed anywhere within the trained range, and could even do so on the fly. In principle, networks did not have to find this unified solution. In practice, training on eight speeds was sufficient to always produce it. This is not necessarily

expected. For example in *Sussillo et al., 2015*, solutions were realistic only when multiple regularization terms encouraged dynamical smoothness. In contrast, for the present task, the stacked-elliptical structure consistently emerged regardless of whether we applied implicit regularization by training with noise. Presumably, the continuously valued input helped make the unified solution natural, such that optimization always found it.

The above interpretation is supported by the fact that the unified solution did not emerge if networks received unrelated inputs (that is, activation of a different input vector) for each speed. Those networks adopted distinct trajectories for each speed (and thus low tangling) but lacked any clear relationship between adjacent speeds (*Figure 3—figure supplement 2c,d*). Despite some disadvantages, in principle the nervous system could use a disjoint solution. Indeed, in some tasks we have observed distinct condition-specific solutions even when not obviously necessary (*Trautmann et al., 2022*). To explore the dependence of the solution on inputs, we also simulated networks that received a different plausible input: simple rhythmic commands (two sinusoids in quadrature) to which networks had to phase-lock their output. Clear orderly stacking with speed was prominent in some networks but not others (*Figure 3—figure supplement 2a,b*). A likely reason for this solution variability is that rhythmic-input-receiving networks had two 'choices'. First, they could use the same stacked-elliptical solution, and simply phase-lock that solution to their inputs. Second, they could adopt solutions with less-prominent stacking (e.g., they could rely primarily on 'tilting' into new dimensions, a strategy we discuss further in a subsequent section). In summary, the stacked-elliptical solution is not inevitable. There were some modeling choices for which the stacked-elliptical solution always occurred (the graded speed-specifying input), some for which it often occurred (simple rhythmic inputs) and some for which it never occurred (distinct speed-specifying inputs). Thus, there is no guarantee that the empirical data will display a stacked-elliptical structure, both because not all networks did, and because the network-based perspective might itself be incorrect.

Yet, while not inevitable, stacked-elliptical structure emerges as a strong prediction of the network perspective under two assumptions. First, that the brain wishes to employ a solution that affords continuity across speeds. Second, that motor cortex is a central participant in the dynamics that generate both the rhythm and the outgoing commands. Networks that embodied these assumptions always displayed the stacked-elliptical structure. As we will explore in a later section, this remained true under a variety of assumptions regarding potential sensory feedback. An important subtlety is that it was not the case that all networks with stacked elliptical structure were performing the identical computation. The flow-field could differ in its stability, and of course network output depended on the monkey / cycling direction. The stacked-elliptical structure is thus not evidence for a specific computation, but rather constitutes a motif that is observed consistently because it is beneficial regardless of the exact computation. This consistency yields straightforward predictions. Qualitatively, empirical neural trajectories should be dominated by elliptical structure within each speed, and separated by a translation across speeds. Quantitatively, this organization should yield trajectory tangling that is low within individual-speed trajectories, and remains low across speeds.

## Neural trajectories are elliptical

We first consider the simplest prediction: the dominant structure of the empirical neural population trajectory, for every speed, should be elliptical regardless of the structure of muscle activity. Neural population activity, for a given time and cycling speed, was a vector containing the trial-averaged response of every sequentially recorded neuron for that time and speed. Muscle population activity was defined analogously. The consistency of behavior (*Figure 1b*) made it reasonable to combine trial-averaged responses from sequentially recorded neurons (and muscles) into a unified population response. To identify the dominant shape of the trajectory for a given speed, we applied PCA to the population response for that condition only (that is, for only that speed and direction) and plotted the projection onto the first two PCs. This approach differs from the typical approach of applying PCA across multiple conditions (as was done in *Figure 3*) and allows us to focus on within-condition structure while ignoring (for now) across-condition structure.

Neural trajectories were approximately elliptical for every condition: all speeds, both cycling directions, and both monkeys (*Figure 4*). In contrast, muscle trajectories exhibited a variety of shapes across speeds and directions, reflecting the different patterns of muscle activity necessary to generate the differing movements. Neural trajectories were more elliptical than muscle trajectories for every

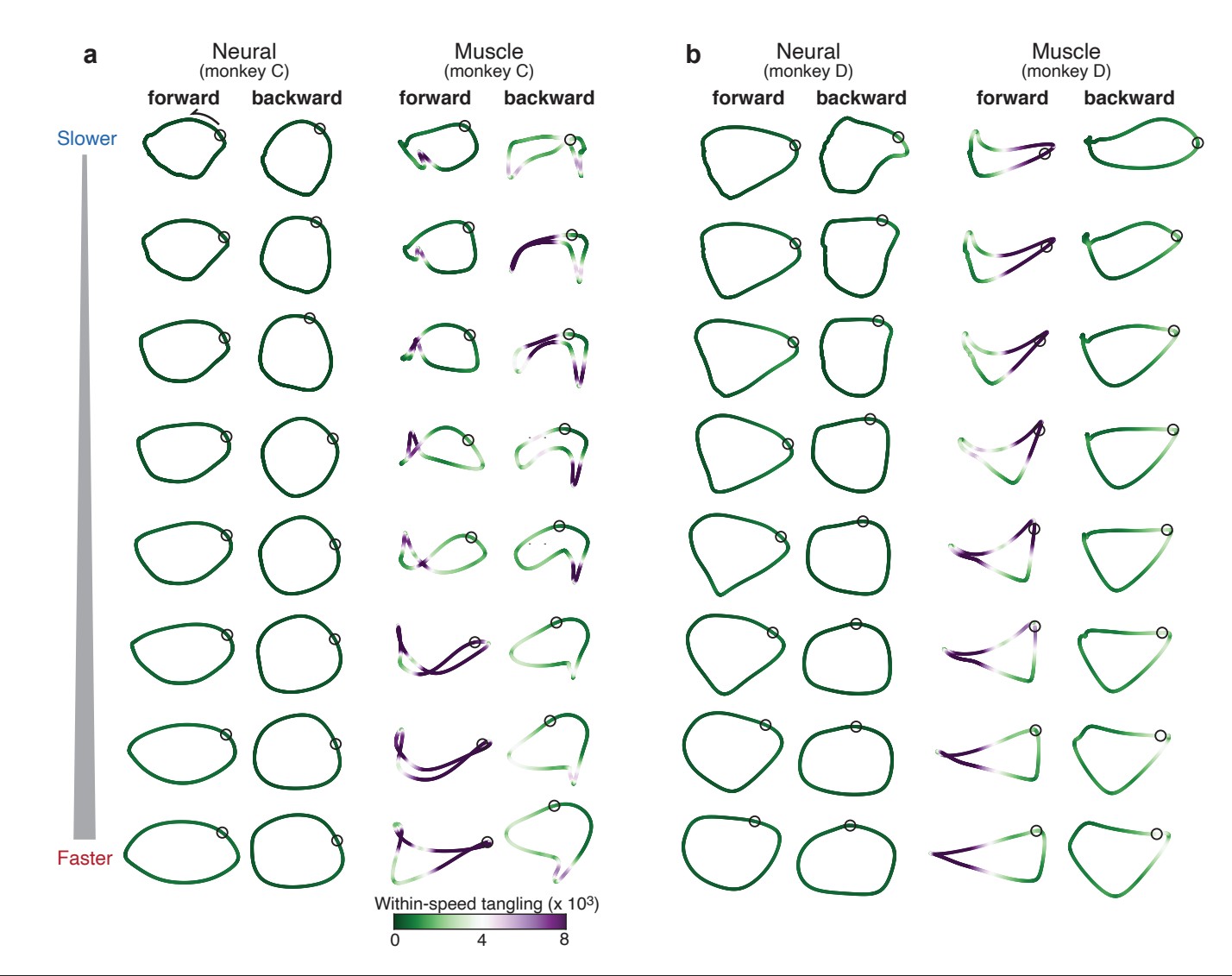

**Figure 4.** Comparison of individual-speed neural trajectories and muscle trajectories. (**a**) Neural and muscle trajectories of monkey C. Trajectories were created by projecting the population response into the top two PCs, with PCs computed separately for each 'condition': that is, each speed bin and cycling direction. Each point within a trajectory corresponds to one time during the cycle for that condition. The time at which the pedal is at the top position is indicated by a black circle. Trajectories are shaded according to the instantaneous tangling value Q(t), computed for these two PCs and entirely within-speed (i.e. only tangling with other points within the same trajectory was considered). (**b**) Same analysis for monkey D.

The online version of this article includes the following figure supplement(s) for figure 4:

**Figure supplement 1.** Same as *Figure 4*, but trajectory tangling was computed in the top six PCs.

condition. This difference was significant for both monkeys (p<0.0001; paired, one-sided Wilcoxon signed rank test comparing $R^2$ values when fitting with an ellipse).

The network perspective predicts that neural trajectory tangling should remain low even if muscle trajectory tangling becomes high. The trajectories in *Figure 4* are colored according to their tangling at every time point, computed as:

$$Q_s\left(t\right) = \max_{t'} \frac{\|\dot{x}_s\left(t\right) - \dot{x}_s\left(t'\right)\|_2^2}{\|x_s\left(t\right) - x_s\left(t'\right)\|_2^2 + \epsilon}$$

(*Equation 1*) where $Q_s\left(t\right)$ is the tangling at time $t$ for speed $s$ and $x_s\left(t\right)$ is the two-dimensional neural or muscle state. To focus on within-trajectory tangling, $t'$ indexes across times within the same cycling

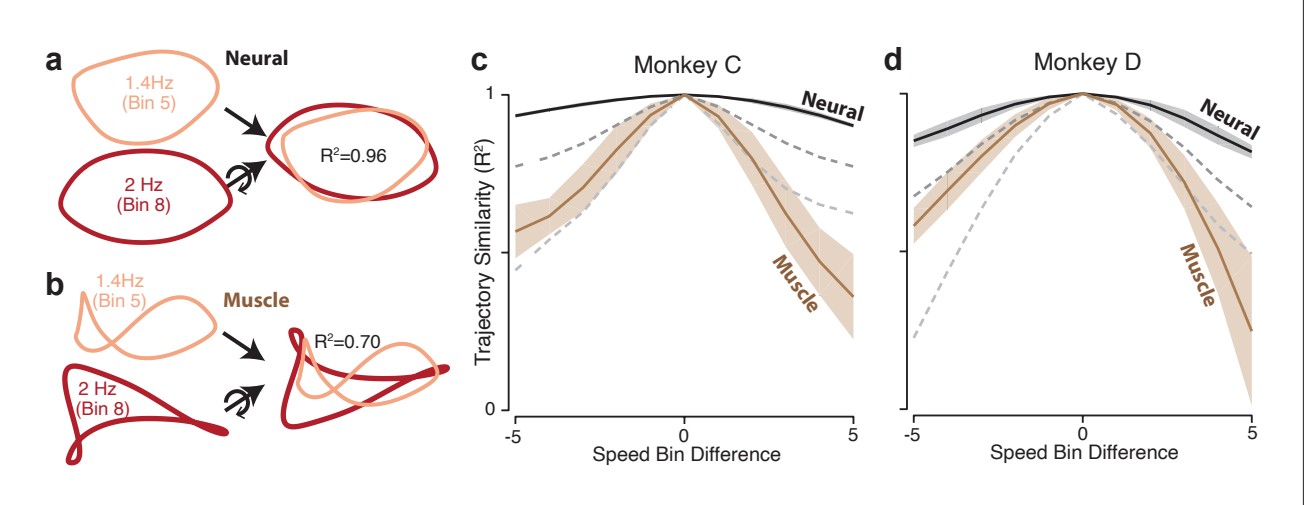

**Figure 5.** Path similarity between trajectories at different speeds. (**a**) Method for calculating neural path similarity. Comparison involved two-dimensional neural trajectories (projected onto PCs 1 and 2) for different speed bins within the same cycling direction. One speed bin (here 1.4 Hz) was chosen as the 'reference trajectory'. Its trajectory path was compared with the trajectory path for each other speed bin (2 Hz is shown for illustration). Comparison of trajectory similarity (computation of $R^2$) followed reflection and rotation, selected to maximize similarity. (**b**) Same as panel (**a**) but for muscle data. (**c**) Average path similarity as a function of speed-bin difference between the reference trajectory and the trajectory with which it is compared. For example, for a speed-bin difference of two, the plotted $R^2$ averages across all comparisons where the comparison trajectory differed from the reference trajectory by two speed bins. Shaded envelopes indicate the SEM. As noted above, the primary analyses compared paths in the top two PCs. For the neural data, we also compared paths in PCs 3 and 4 (dark dashed gray line) and PCs 5 and 6 (light dashed gray line). (**d**) Same as (**c**) but for monkey D.

direction and speed. Muscle tangling was often high (*purple*) due to tightly kinked regions, trajectory crossings, or nearby segments traveling in opposing directions. Neural trajectory tangling was much lower. This was true for every condition and both monkeys (paired, one-tailed t-test; p<0.001 for every comparison). This difference relates straightforwardly to the structure visible in the top two PCs; the effect is present when analyzing only those two PCs and remains similar when more PCs are considered (*Figure 4—figure supplement 1*). There is no straightforward relationship between high versus low trajectory tangling and high versus low dimensionality (*Russo et al., 2018*). Instead, the degree of tangling depends primarily on the structure of trajectories in high-variance dimensions. Neural trajectories have lower tangling because their dominant structure is elliptical.

## Neural trajectories are similar across speeds

The dominant structure of neural activity was surprisingly insensitive to the details of muscle activity; neural trajectories had a similar shape for every speed, even though muscle trajectories did not. Ellipse eccentricity changed modestly across speeds but there was no strong or systematic tendency to elongate at higher speeds (for comparison, an approximately threefold elongation would be expected if one axis encoded cartesian velocity). We quantified trajectory-shape similarity among speeds within the same cycling direction (that is, within a column in *Figure 4*). Because PCA was applied per speed, trajectories have different bases, yet one can still ask whether they share the same shape. To do so, we take two trajectories, $x_{ref}(t)$ and $x_i(t)$, apply the rigid rotation that maximizes their similarity, and measure that similarity. This procedure was performed for neural and muscle trajectories (*Figure 5a and b*). Similarity was defined as $R^2(x_{ref}, x_i)$, the variance in $x_i(t)$ accounted for by $x_{ref}(t)$, with no scaling or offset allowed. We defined $R_k^2$ as the average $R^2(x_{ref}, x_i)$ over all situations where $x_{ref}(t)$ and $x_i(t)$ were separated by $k$ speed bins.

$R_k^2$ is by definition unity for $k = 0$, and is expected to decline for larger values of $k$. That decline was modest for neural trajectories and steeper for muscle trajectories (*Figure 5c and d*). This confirms that neural trajectories had a consistent shape in the dominant two dimensions, while muscle-trajectory shapes were speed-specific. Yet if neural trajectories encode commands that generate muscle activity, there must be dimensions where neural trajectories have different shapes across speeds. This was indeed the case. We repeated the analysis using neural trajectories that were projections onto PCs three and four (top dashed gray curve) or five and six (bottom dashed gray curve). These aspects of

the neural population response were less similar across speeds. For example, when considering neural PCs five and six, the decline in $R_k^2$ was comparable to that for the first two muscle PCs. This agrees with a fundamental feature of network solutions: output-encoding signals are 'pushed' into higher dimensions, allowing tangling to remain low in the dominant dimensions.

## Neural population trajectories across speeds

Network solutions generated two predictions regarding the organization of neural activity across speeds. Qualitatively, trajectories should be separated by a translation across speeds. Quantitatively, that separation should be large and consistent enough to yield low overall tangling (something not guaranteed only from average separation). We evaluated the first prediction by applying PCA jointly across all speeds for a given cycling direction. Unlike in the speed-specific projections above, a unified two-dimensional projection does not perfectly capture the elliptical trajectories because they unfold in somewhat speed-specific dimensions (as will be explored further below) and are thus modestly distorted when all projected into the same two dimensions. Nevertheless, the first two PCs did a reasonable job capturing the dominant elliptical structure, which was lacking in the muscle trajectories (*Figure 6a*). Employing three PCs revealed the predicted translation separating individual-speed neural trajectories (*Figure 6b*, *left*). Considered in three PCs, neural and muscle trajectories (*Figure 6b*, *right*) differed in much the same way as network and muscle trajectories. Neural trajectories were stacked ellipses, while muscle trajectories had a less clear organization.

There is no particular reason that the 'speed axis', the dimension that best captures stacking across speeds, should align with the third PC. For monkey C, the third PC naturally revealed stacking, but that feature was clearer (*Figure 6c*) when we identified the speed axis directly. For monkey D, stacking was not particularly prevalent in the third PC, but was quite apparent when the speed axis was found directly (*Figure 6—figure supplement 1*). The speed axis was defined as the dimension where the mean neural state (averaged across times within a trajectory) provided the best decode of mean angular velocity (Materials and methods). In every plot, all dimensions share the same scale. The speed axis captured 4.94% and 4.37% of the total variance (Monkey C and D). This was only modestly less than the variance captured by the third PC (5.34% and 8.57%) which defines the maximum variance that could have been captured by the speed axis given that the first two PCs captured the dominant elliptical structure.

When considered in more than two dimensions, neural trajectories could depart from planar ellipses, sometimes tracing a pringle-like (hyperbolic paraboloid) shape. The same was true for many networks (e.g. *Figure 3e*). In principle such departures could increase trajectory tangling between speeds – neighboring trajectories could come close at moments where local path direction differs. Yet for both neurons and networks, this did not occur because neighboring trajectories tended to depart from planar in similar ways. This suggests agreement with a prediction derived from the networks: individual-trajectory shapes, combined with between-trajectory separation, should prevent across-speed tangling. To test this, we employed *Equation 1* but with $t'$ indexing across times in all other speeds (and not within the same speed). This complements the within-speed tangling analysis above. Tangling was computed in 12 dimensions as these captured most of the variance within and between conditions (results were extremely similar if more dimensions were employed).

Neural trajectories had consistently low across-speed tangling (*Figure 6c*, *left*, *green shading* indicates low tangling). In contrast, muscle trajectories often displayed high across-speed tangling (*Figure 6c*, *right*, *purple shading* indicates high tangling). This difference was pronounced for both monkey C (*Figure 6c*) and D (*Figure 6—figure supplement 1c*). Muscle trajectories did display a rough 'speed axis' (more pronounced for monkey C, less so for monkey D) in the sense that the trajectory mean differed across speeds. Yet muscle-trajectory tangling was still high. This underscores the specificity of the network-derived prediction: neural trajectories should be separated not just on average, but in a consistent manner that maintains low tangling. This was true of the neural trajectories but not the muscle trajectories.

## Low global tangling depends upon inter-speed separation

As documented above, neural trajectory tangling was low when considering within-trajectory comparisons (*Figure 4*) and also when considering between-trajectory comparisons (*Figure 6c*). Muscle trajectory tangling was high in both cases. A reasonable inference is thus that global trajectory tangling

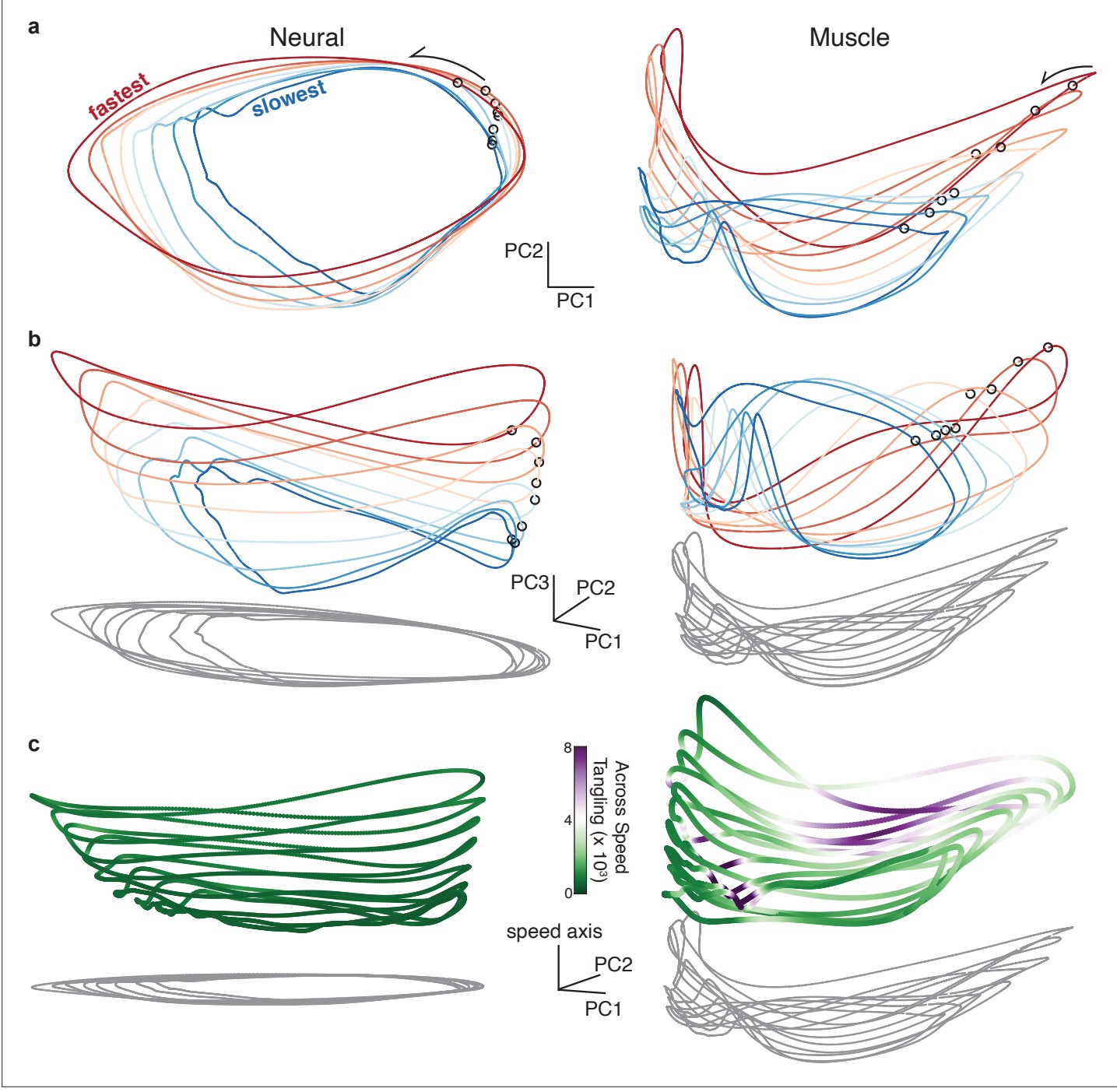

**Figure 6.** Neural and muscle population trajectories across speeds, with trajectories for every speed projected into a common basis. All data are for monkey C, forward condition. (**a**) Neural and muscle activity projected onto global PCs 1 and 2: the dominant dimensions when PCA was applied across all speeds for that cycling direction (**b**) Same as (**a**) but for PCs 1, 2, and 3. (**c**) Same as (**a**) but for PCs 1, 2, and the 'speed axis' (see text for how this was found). Trajectories in (**c**) are colored according to instantaneous tangling values, considering only tangling across conditions and ignoring any within-condition tangling. Tangling was computed in a twelve-dimensional PC space, computed by applying PCA to the data for all speeds for a given direction.

The online version of this article includes the following figure supplement(s) for figure 6:

**Figure supplement 1.** Same as *Figure 6* but for forward cycling for monkey D.

– employing all comparisons within and between trajectories – should be much lower for neurons versus muscles. This was indeed the case (*Figure 7*, black dots). Each dot represents global trajectory tangling at one time for one speed, computed by allowing $t'$ in **equation 1** to index across all times for all speeds. There were many moments when global trajectory tangling became high for the muscle populations. In contrast, global trajectory tangling never became high for the neural populations. The difference between neural and muscle trajectory tangling was not due to differences in intrinsic dimensionality; it was present whether we employed a matched number of dimensions for neural and muscle data (as in *Figure 7*), employed as many dimensions as needed to capture >95% of the variance (typically more dimensions for the neural data), or employed no dimensionality reduction and simply computed tangling in the full-dimensional space (which yielded nearly identical results but was time-consuming).

For the simulated networks, global tangling was only slightly higher than within-speed tangling (*Figure 7—figure supplement 1*); global tangling was kept low by the separation of trajectories along the speed axis described above. For the neural data, it was similarly true that tangling was low when computed purely within speed (*orange circles* and *bars*) and rose very little when computed globally (*black circles* and *bars*). We used two methods to confirm that the lack of rise was due to trajectory separation between speeds. First, we constructed an artificially rescaled population response by taking the empirical neural trajectory at a reference speed and constructing a trajectory for every other speed by rescaling time. We 'recorded' single-neuron responses (spikes generated via a Poisson process) and analyzed the artificial population in the same way as the recorded neural data. This procedure yields trajectories that overlap but evolve at different speeds, increasing tangling (*dark gray bars*). Second, we took the empirical neural population response and removed both the separation between trajectory means and any separation created by tilting of trajectories into different dimensions (a feature documented below). This also led to an increase in global tangling (*light gray bars*). Thus, for both the cortical and network trajectories, the dominant elliptical structure ensures low within-speed tangling, and separation between trajectories maintains low global tangling.

## Organization of trajectory separation

Low tangling requires not just overall separation (different trajectory means) but consistent separation at all points along neighboring trajectories. The low global neural-trajectory tangling argues that such separation is present, but can it be confirmed directly? We calculated the distance between trajectories as a function of $\theta$, the phase of the physical movement. We chose a middle speed (speed bin 4) as a 'reference' trajectory, and computed the Euclidean distance to all other trajectories at every phase (*Figure 8a*). To link with the analysis in *Figure 4c*, we also computed the distance between trajectory means. The distance between trajectory means behaved as anticipated: it grew monotonically as a function of the difference in speed from the reference trajectory (*Figure 8b*). A similar pattern was observed when distance was considered as a function of $\theta$ (*Figure 8c*). As speeds became more different from the reference speed, the corresponding trajectories became more distant. Most critically, there was never a value of $\theta$ where the distance became very small. Thus, no other trajectory, for any other speed, came 'close' to the reference trajectory. Moreover, distance generally grew monotonically with increasing differences in speed.

The above remained true regardless of the speed chosen for the reference trajectory. This is worth noting because, in *Figure 8c*, traces corresponding to speeds distant from the reference speed occasionally touch (e.g. the blue traces at bottom). This could imply either that those trajectories lack separation, or simply that they are equidistant from the reference trajectory but in somewhat different directions. These possibilities can be distinguished by repeating the analysis with different speeds as the reference trajectory. For example, if we chose speed-bin 2 (corresponding to the traces second from the bottom in *Figure 8c*) as the reference trajectory, it was well separated from the trajectory for speed-bin 1 and 3 at all phases (*Figure 8—figure supplement 1a*). This organization resembles that observed during a cognitive task where intervals must be internally timed (*Remington et al., 2018*), indicating that the geometry of neural activity can reflect the essence of a computational problem (adjusting the rate at which something is accomplished) across different tasks (cognitive versus motor) and brain areas (medial prefrontal versus motor cortex).

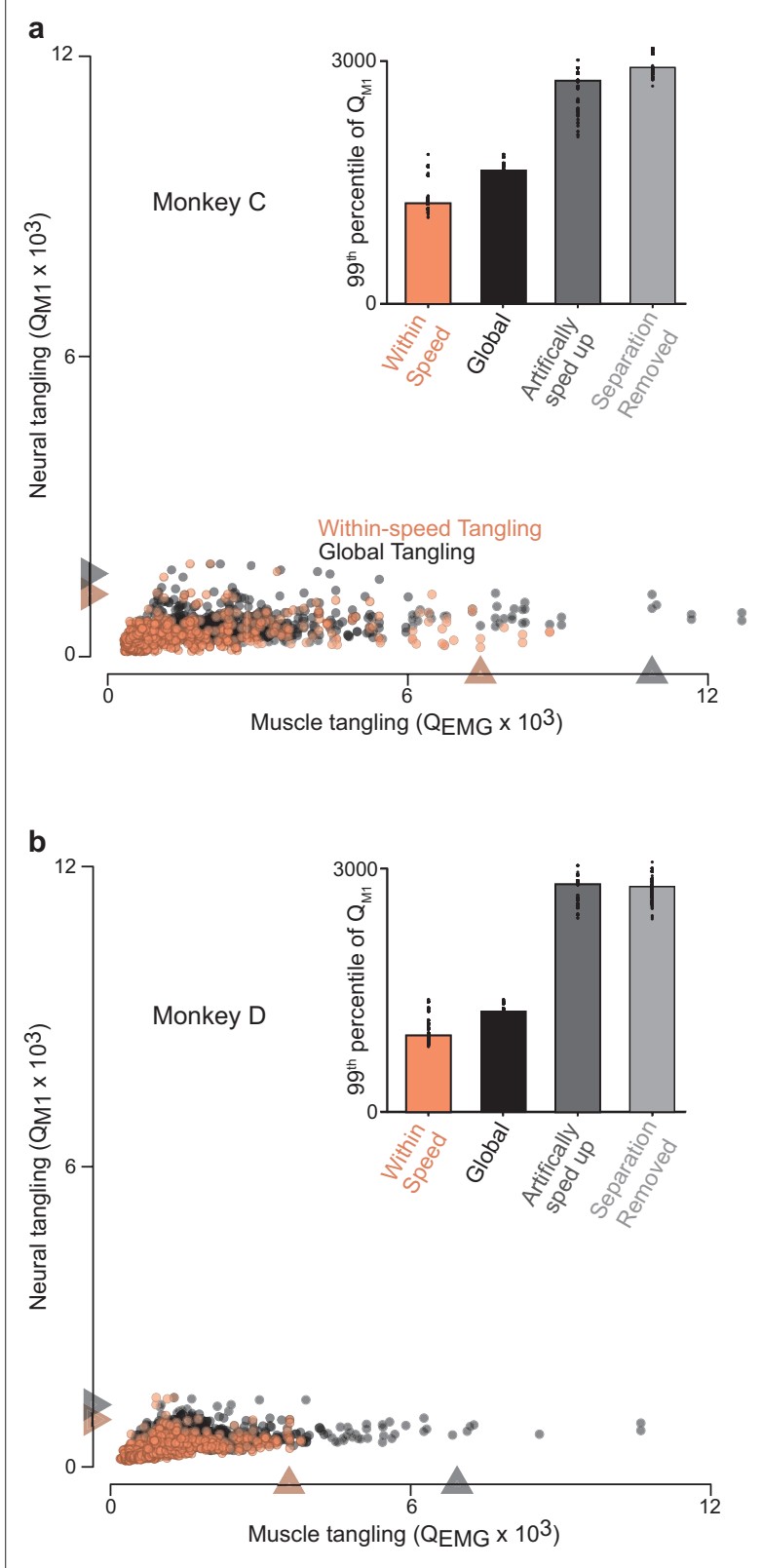

**Figure 7.** Trajectory tangling values for purely within-speed comparisons and global comparisons (within and across all speeds for a given direction). (**a**) Scatterplot of neural-trajectory tangling versus muscle-trajectory tangling (monkey C) for within-speed tangling (*orange*) and global tangling (*black*). Each point shows tangling for one moment (one time during one speed and one direction). Points are shown for all times during movement

*Figure 7 continued on next page*

*Figure 7 continued*

(sampled every 25ms) for all sixteen conditions. *Gray / orange triangles* indicate 99th percentile tangling. All tangling values (both within-speed and global) were computed in the same twelve-dimensional global space found by applying PCA to all the data (that is, all speeds) for that cycling direction. The difference in the within-speed versus global computations simply involved whether the computation of tangling included only the trajectory for the speed containing a given state, or included all other trajectories. Note that, by construction, global tangling must be at least as high as within-speed tangling. **Inset** plots 99th percentile tangling values for within-speed tangling and global tangling. Also shown are values obtained by manipulating the empirical data to create new data without separation between speeds. For the 'artificially sped up' manipulation, all trajectories were identical in path but evolved according to their original rates. For the 'separation removed' manipulation, all trajectories were given the same mean and forced to unfold in the same dimensions. *Black dots* show 99th percentile tangling values for bootstrapped data (Materials and methods). (**b**) Same as (**a**) but for Monkey D.

The online version of this article includes the following figure supplement(s) for figure 7:

**Figure supplement 1.** Similar to insets in *Figure 7*, but 'neural' tangling assesses the population response of an example RNN rather than recorded population activity.

## Trajectories separate into different dimensions

All network solutions exhibited one additional feature: the plane that best captured the elliptical trajectory for the slowest speed was moderately different from that for the fastest speed, providing an additional form of separation between network trajectories (*Figure 8—figure supplement 2a,b*). Do empirical neural trajectories exhibit this same feature? One observation above suggests so: elliptical trajectories were more cleanly captured when PCA was applied to each speed separately (*Figure 4*) rather than all speeds together (*Figure 6a*). This suggests that the elliptical trajectories for different speeds are not perfectly parallel to one another (if they were, looking down on the stack should reveal each and every elliptical trajectory perfectly).

To test this prediction directly, we computed the top two PCs from the slowest-speed trajectory, which captured >75% of the variance for that trajectory. We fixed these PCs and asked how much variance they explained for the other trajectories. Variance explained declined monotonically as speed increased, reaching a minimum of ~40% for the fastest speed. The same effect was observed in reverse if we computed the two PCs from the fastest trajectory and then considered slower trajectories (*Figure 8d and e*). This effect held even if we considered the top ten PCs (*Figure 8—figure supplement 2c,d*). Thus, trajectories unfolded in dimensions that overlapped but became progressively more different for larger differences in speed.

These findings are consistent with suggestions regarding how a circuit can generate multiple behaviors (*Briggman and Kristan, 2008*) and with empirical and network solutions across distinct behaviors such as forward and backward cycling (*Russo et al., 2018*) or cycling with one arm versus the other (*Ames and Churchland, 2019*). The present results indicate that the same principle – 'tilting' into different dimensions to alter motor output – is operative when continuously adjusting a specific behavior (also see [*Sabatini and Kaufman, 2021*]). Yet while the separation across individual-speed trajectories was sufficient to aid low tangling, it was modest enough to allow solutions to remain related. For example, the top PCs defined during the fastest speed still captured considerable variance at the slowest speed. Network simulations (see above) show both that this is a reasonable strategy and also that it isn't inevitable; for some types of inputs, solutions can switch to completely different dimensions across speeds. The presence of modest tilting likely reflects a balance between tilting enough to alter the computation while still maintaining continuity of solutions.

## Generality of the network solution

Network optimization consistently found solutions whose structure resembled that of the empirical data, even though networks were unrealistic in many specifics. Networks lacked realistic cell-types; indeed their units did not even spike. Nevertheless, the dominant population-level factors were similar to factors estimated from real spiking neurons. This presumably occurs because the advantages of low trajectory tangling apply regardless of implementation. Recurrent networks with rate-based units intrinsically employ continuous-valued dynamics, but continuous-valued dynamical systems can also be instantiated by spiking networks (*Boerlin et al., 2013*; *Eliasmith, 2005*). Indeed, the same factor-level dynamics can be instantiated by both rate-based and spiking networks (*DePasquale et al.,*

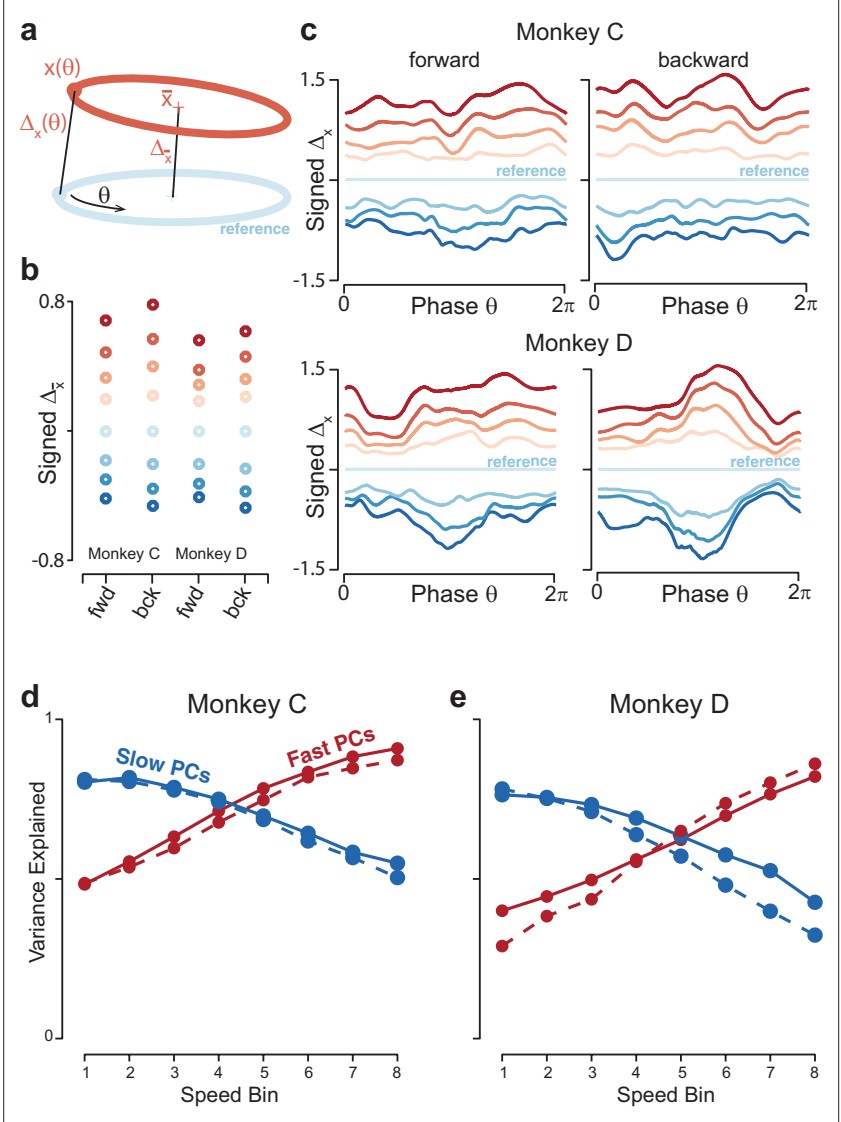

**Figure 8.** Quantification of separation between neural trajectories at different speeds. (**a**) Illustration of the method for calculating distance as a function of phase. One speed bin was chosen to provide the reference trajectory. For the present analysis this was always speed bin 4, and the cartoon reference trajectory (*light blue*) is thus colored accordingly. We then took a trajectory from another speed bin (*orange*). We swept the phase, $\theta$, of the reference trajectory, and for each phase computed the distance to the nearest point of the other trajectory. We also computed the distance between the trajectory means. (**b**) Distance between trajectory means, for each monkey and cycling direction. Speed bin 4 is the reference trajectory, and thus distance is zero. (**c**) Phase-dependent distance of the reference trajectory from each of the other trajectories. (**d**) Quantification of the degree to which elliptical trajectories unfold in the same set of dimensions across speeds. Slow PCs were the first two PCs based on speed bin 1, and fast PCs were the same based on speed bin 8. For each set of PCs, we computed the proportion of variance explained for each of the other speed bins. The analysis was performed separately for forward and backward cycling (*solid* and *dashed lines*, respectively). This analysis is repeated, using the top ten PCs, in *Figure 8—figure supplement 2C, d*. Data are for monkey C. (**e**) Same for monkey D.

The online version of this article includes the following figure supplement(s) for figure 8:

**Figure supplement 1.** Similar to *Figure 8c*, but using different speed bins for the reference trajectory.

**Figure supplement 2.** Analyses related to those in *Figure 8d and e*.

*2016*), including networks with separate excitatory and inhibitory sub-populations. In such networks, dynamics display low-tangling at the level of the factors (though not at the level of spiking).

Considerable generality of solutions was also seen with respect to the particulars of network outputs. We trained networks to fit four sets of muscle activity (two monkeys and cycling directions) that differed considerably. The dominant stacked-elliptical structure persisted across networks (*Figure 9—figure supplement 1*). This matters because the modeling assumption that outgoing cortical commands are isomorphic with muscle activity is only an approximation. Meaningful transformations presumably occur between cortical outputs and motoneuron activity (*Shalit et al., 2011*; *Albert et al., 2020*), and motoneurons receive other sources of drive. One thus wishes to know that network predictions are relatively insensitive to assumptions regarding the exact output, and this was indeed the case. This insensitivity relates to the form of the solution. The dominant elliptical trajectory both ensures low tangling and can provide the fundamental output frequency. All other aspects of the output are built from smaller signals in dimensions orthogonal to the dominant trajectory. Thus, the stacked-elliptical solution does not instantiate the full computation. Rather, it is a scaffolding upon which a computation, i.e. an input-output relationship, can be built. The structure of the scaffolding reflects broad needs, such as the fact that motor output varies with phase and repeats once per cycle, and is thus largely independent of the details of the input-output relationship. That input-output relationship depends upon activity in lower-variance dimensions. For example, in our networks, for each muscle there exists a dimension where network activity encodes that muscle's activity. These output-encoding signals are 'representational' in the sense that they have a consistent relationship with a concrete quantity. In contrast, the dominant stacked-elliptical structure exists to ensure a low-tangled scaffold and has no straightforward representational interpretation.

Given that stacked-elliptical structure can instantiate a variety of input-output relationships, a reasonable question is whether networks continue to adopt the stacked-elliptical solution if, like motor cortex, they receive continuously evolving sensory feedback. We found that they did. Networks exhibited the stacked-elliptical structure for a variety of forms of feedback (*Figure 9b and c*, *top rows*), consistent with prior results (*Sussillo et al., 2015*). This relates to the observation that 'expected' sensory feedback (i.e. feedback that is consistent across trials) simply becomes part of the overall network dynamics (*Perich et al., 2020*). Network solutions remained realistic unless the influence of feedback became so strong that it dominated network dynamics (*Figure 9b and c*, *bottom rows*).

We did not attempt to simulate feedback control that takes into account unpredictable sensory inputs and produces appropriate corrections (*Stavisky et al., 2017*; *Pruszynski and Scott, 2012*; *Pruszynski et al., 2011*; *Pruszynski et al., 2014*). However, there is no conflict between the need for feedback control and the general form of the solution observed in both networks and cortex. Consider an arbitrary feedback control policy:

$$z = g_c\left(t, u_f\right)$$

where $u_f$ is time-varying sensory input arriving in cortex and $z$ is a vector of outgoing commands. The networks we trained all embody special cases where $u_f$ is either zero (most simulations) or predictable (*Figure 9b and c*) and the particulars of $z$ vary with monkey and cycling direction. Stacked-elliptical structure was appropriate in all these cases and would likely continue to be an appropriate scaffolding for control policies with greater realism, although this remains to be explored.

## Discussion

The ability of animals to adjust movement speed provides a test for the network-dynamics perspective: can it explain the dominant structure of neural responses? We found that it could, at both the single-neuron and population levels. Single-neuron responses were better explained when regressing against network factors rather than muscle factors. This is notable because even qualitatively accurate network solutions are not guaranteed to provide quantitatively useful factors, especially when competing against muscle factors that naturally share features with neural responses. Yet network factors essentially always provided some improvement, and frequently provided large improvements. More deeply, network solutions anticipated the 'stacked-elliptical-trajectory' organization of neural population trajectories. That organization explained the seeming paradox that neural trajectories are elliptical, independent of muscle-trajectory shape. This disconnect would, from a traditional

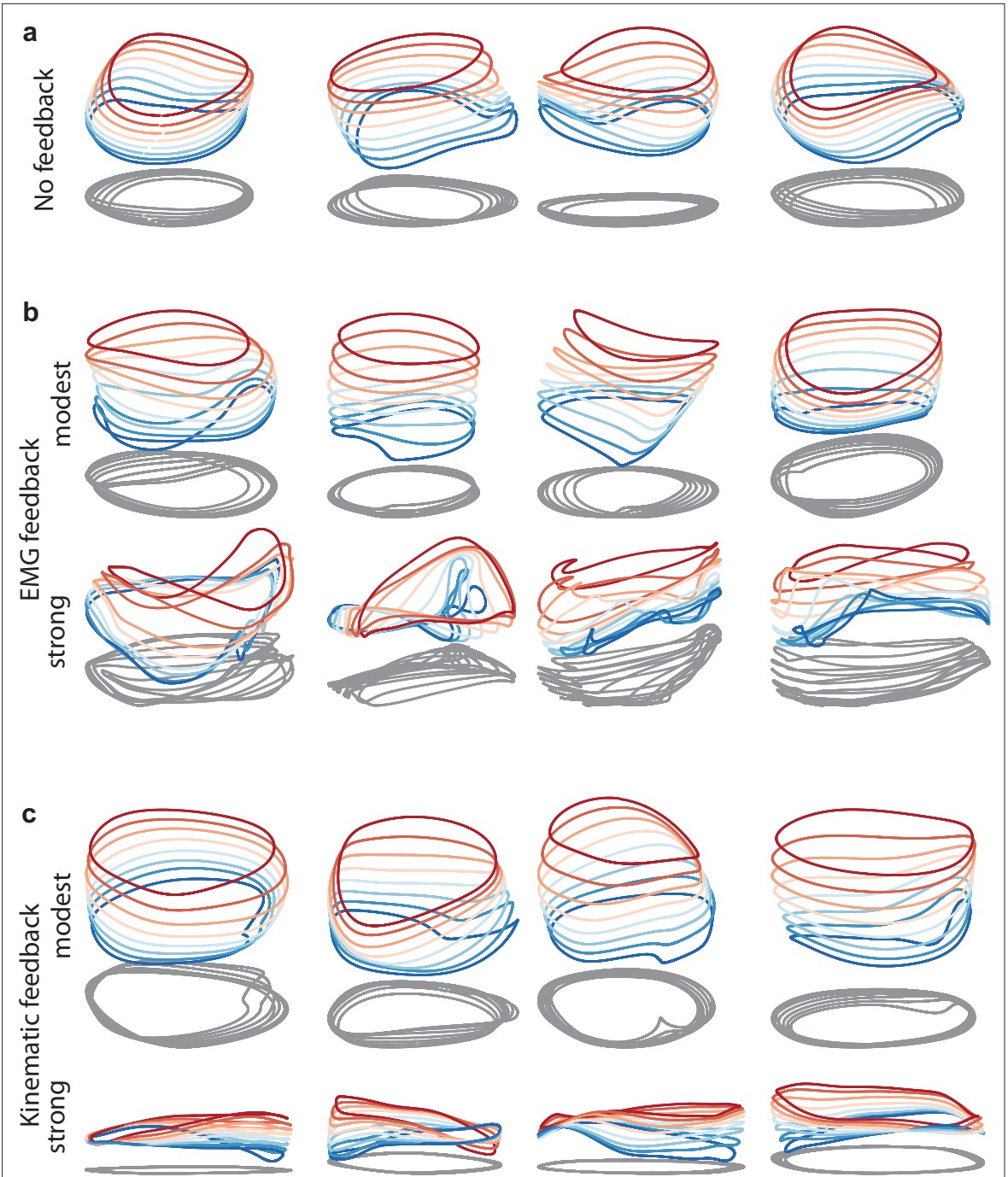

**Figure 9.** Network solutions with and without simulated sensory feedback. Network inputs and outputs were as in *Figure 3*, except some networks were supplied with an additional input conveying delayed sensory feedback related to prior motor output. All examples were trained to produce outputs based on the muscle activity of Monkey C during forward cycling. Training occurred in the presence of all relevant inputs (including feedback if present). (**a**) Networks with no explicitly simulated sensory feedback (similar to *Figure 3d*). (**b**) Networks supplied with delayed EMG feedback (PCs 1–6 of muscle activity; 60ms delay) at a gain of 0.1 (top) and a gain of 10 (bottom). (**c**) Networks supplied with delayed kinematic feedback (horizontal and vertical position and velocity signals; 60ms delay) at a gain of 0.1 gain (top), and a gain of 10 (bottom).

The online version of this article includes the following figure supplement(s) for figure 9:

**Figure supplement 1.** Same as in *Figures 9a and 3d*, but with additional example networks for the other cycling direction and the other monkey.

perspective, have argued against a tight relationship between neural and muscle activity. In fact, the opposite is true: this property is expected of a network that generates muscle commands (or more broadly, descending commands that will shortly be converted into muscle activity).

Network solutions and neural activity were also similar at a more abstract level. An emerging approach is to consider geometric properties above the level of individual-task responses. Certain classes of computation require or imply certain geometric properties (*Russo et al., 2018*; *Russo et al., 2020*; *Bernardi et al., 2018*; *Sohn et al., 2019*; *DiCarlo et al., 2012*; *Hénaff et al., 2019*). The level of geometry illuminates why neural activity likely employs stacked-elliptical structure: such structure maintains both low trajectory tangling and solution continuity. This perspective reveals commonalities among seemingly disparate features. Trajectories were separated by both a translation and by occupation of moderately different dimensions. These features seem unrelated at the single-neuron level: the translation produces firing-rate profiles with different offsets while the use of different dimensions yields speed-specific neuron-neuron correlations. Yet both help maintain low global tangling.

The desirability of low tangling holds across a broad range of situations (*Russo et al., 2018*). Consistent with this, we observed stacked-elliptical structure in networks that received only speed-specifying commands, and in many of the networks that received rhythmic forcing inputs. Thus, the empirical population response is consistent with motor cortex receiving a variety of possible input commands from higher motor areas: a graded speed-specifying command, phase-instructing rhythmic commands, or both. The stacked-elliptical solution was also present in networks that received a variety of forms of simulated periodic feedback. We did not simulate networks that had to respond to unpredictable feedback, but we predict the dominant structure would remain similar – the need to maintain low tangling still holds. Yet the dominant stacked-elliptical structure was not an inevitable aspect of network solutions; networks used other strategies if different speeds were instructed by unrelated inputs, or if simulated feedback was so strong that it dominated network activity.

This second observation highlights an important subtlety. The dynamics shaping motor cortex population trajectories are widely presumed to reflect multiple forms of recurrence (*Churchland et al., 2012*): intracortical, multi-area (*Middleton and Strick, 2000*; *Wang et al., 2018*; *Guo et al., 2017*; *Sauerbrei et al., 2020*) and sensory reafference (*Lillicrap and Scott, 2013*; *Pruszynski and Scott, 2012*). Both conceptually (*Perich et al., 2020*) and in network models (*Sussillo et al., 2015*), predictable sensory feedback becomes one component supporting the overall dynamics. Taken to an extreme, this might suggest that sensory feedback is the primary source of dynamics. Perhaps what appear to be 'neural dynamics' merely reflect incoming sensory feedback mixed with outgoing commands; a purely feedforward network could convert the former into the latter and might appear to have rich dynamics simply because the arm does (*Kalidindi et al., 2021*). While plausible, this hypothesis strikes us as unlikely. It requires sensory feedback, on its own, to create low-tangled solutions across a broad range of tasks. Yet there exists no established property of sensory signals that can be counted on to do so. If anything the opposite is true: trajectory tangling during cycling is relatively high in somatosensory cortex even at a single speed (*Russo et al., 2018*). The hypothesis of purely sensory-feedback-based dynamics is also unlikely because population dynamics begin unfolding before movement begins (*Churchland et al., 2012*). To us, the most likely possibility is that internal neural recurrence (intra- and inter-area) is adjusted during learning to ensure that the overall dynamics (which will incorporate sensory feedback) provide good low-tangled solutions for each task. This would mirror what we observed in networks: when present at reasonable levels, sensory feedback influenced dynamics but did not determine its dominant structure. Instead, the stacked-elliptical solution emerged because it was a 'good' solution that optimization found by shaping recurrent connectivity.

The presence of low tangling in motor cortex, across many tasks, supports the hypothesis that dynamics rely in part on internal sources of recurrence that are adjusted to provide noise-robust solutions. Yet even in a network that relies on strong dynamics, low trajectory tangling is neither always desirable nor always expected. There can be moments where activity must be dominated by 'unexpected' inputs. Consider the standard delayed-reach task. Upon target onset, the neural state is driven in different directions depending on target location. This brief moment of high tangling is desirable – one does not want the system's own state to determine its future state because target-specific inputs should do so. In the present study, networks can adjust cycling speed when their input changes, and swift changes would increase tangling. Similar considerations may explain why trajectory

tangling is low (and dynamical fits good) in motor cortex during reaching but not during grasping (*Suresh et al., 2019*). As suggested in that study, grasping may require a more continuous flow of guiding inputs from the rest of the brain.

Even when low tangling is desirable, it is desirable at the level of the full dynamical system, not necessarily within any given brain area. For example, somatosensory cortex exhibits high tangling (*Russo et al., 2018*), as does motor neuron activity, even though both are part of a movement-generating dynamical system. Given this, it is not inevitable that trajectory tangling is low in motor cortex or that activity is well-described by autonomous dynamics. Indeed it is quite surprising; intrinsic motor cortex connectivity is presumably only partially responsible for the overall movement generating dynamics. A likely explanation is that the overall dynamics are often close to fully observable in motor cortex (*Sussillo et al., 2015*; *Seely et al., 2016*) because it forms a hub for multiple forms of internal (*Sauerbrei et al., 2020*; *Middleton and Strick, 2000*; *Wang et al., 2018*; *Guo et al., 2017*) and sensory recurrence (*Pruszynski and Scott, 2012*). If neurons in motor cortex reflect most of the key system-wide state variables, then the motor cortex population response will be describable by $\dot{x} = f(x) + u$ where the $f(x)$ term dominates. Conversely, a large $u$ term would be needed in situations where not all key variables are reflected.

Such considerations may explain why (*Foster et al., 2014*), studying cortical activity during locomotion at different speeds, observed stacked-elliptical structure with far less trajectory separation (<1% of the population variance) which is unlikely to provide enough separation to minimize tangling. This agrees with the finding that speed-based modulation of motor cortex activity during locomotion is minimal (*Armstrong and Drew, 1984*) or modest (*Beloozerova and Sirota, 1993*). The difference between cycling and locomotion may reflect cortex playing a less-central role in the latter. During locomotion, cortical activity may reflect being 'informed' of the spinally generated rhythm for the purpose of generating gait corrections if necessary (*Drew and Marigold, 2015*; *Beloozerova and Sirota, 1993*). If so, cortical trajectories needn't display low tangling as they would be largely input driven.

The observed continuity of neural solutions agrees with the finding that improving a motor skill at one speed generalizes well to other speeds (*Shmuelof et al., 2012*). More broadly, stacked-elliptical structure is likely a common solution to the problem of adjusting the speed of a process or computation. Similar structure has been observed across artificial networks, with multiple architectures, trained to produce idealized periodic signals at different speeds (*Maheswaranathan et al., 2019*). A similar solution is observed in a cognitive task, both for artificial networks and in frontal-cortex population activity (*Remington et al., 2018*). This reinforces that the dominant features of network solutions often reflect the nature of a computational problem rather than the specific input-output function or the particular domain (e.g. motor versus cognitive). Similar trajectory geometries can even serve very different computational purposes. For example, in the supplementary motor area, trajectory cycles are separated during progress within a larger action (e.g., travelling seven cycles and then stopping), producing a helical trajectory. In motor cortex, our present model predicts helical trajectories when speed steadily increases (something we have informally observed during different experiments). The helical solutions in the supplementary motor area and in motor cortex have something high-level in common, yet the nature of the underlying computation is completely different (keeping track of progress, versus specifying speed).

This underscores that certain trajectory 'motifs' are broadly useful, and constrain but do not fully specify the underlying computation. Conversely, seemingly similar computational problems can require different solutions. Even the simple goal of 'speeding up movement' likely requires very different solutions depending on what it means to 'speed up'. For example, generating a particularly fast reach does not involve shorter-duration muscle activity, but requires larger accelerations and decelerations, resulting in muscle activity with more prominent peaks and valleys in the ~2 Hz range. Consistent with this, motor cortex activity exhibits a higher-amplitude ~2 Hz oscillatory pattern during fast reaches (*Churchland et al., 2012*). At the other end of the spectrum, movement sequences are accelerated by reducing the time between preserved events (*Zimnik and Churchland, 2021*). Thus, there is no single strategy for speed-control across all behaviors. However, in all these cases – single reaches, reach sequences, and cycling – the network-dynamics perspective is helpful in comprehending the computational strategies employed to adjust speed.

## Materials and methods

### Experimental apparatus

Monkeys (C and D, two adult male rhesus macaques) were trained to perform a cycling task (*Russo et al., 2018*). Animal protocols were approved by the Columbia University Institutional Animal Care and Use Committee (Protocol number AC-AABE3550). Experiments were controlled and data collected using the Speedgoat Real-time Target Machine. During experiments, monkeys sat in a customized chair with the head restrained via a surgical implant. Stimuli were displayed on a monitor in front of the monkey. A tube dispensed juice rewards. The left arm was loosely restrained using a tube and a cloth sling. With their right arm, monkeys manipulated a pedal-like device. The device consisted of a cylindrical rotating grip (the pedal), attached to a crank-arm, which rotated upon a main axle. That axle was connected to a motor and a rotary encoder that reported angular position with 1/8000 cycle precision. In real time, information about angular position and its derivatives was used to provide virtual mass and viscosity, with the desired forces delivered by the motor. The delay between encoder measurement and force production was 1ms.

Horizontal and vertical hand position were computed based on angular position and the length of the crank-arm (64 mm). To minimize extraneous movement of the wrist, the right wrist rested in a brace attached to the hand pedal. The motion of the pedal was thus almost entirely driven by the shoulder and elbow, with the wrist moving only slightly to maintain a comfortable posture. Wrist movements were monitored via two reflective spheres attached to the brace, which were tracked optically (Polaris system; Northern Digital, Waterloo, Ontario, Canada) and used to calculate wrist angle. The small wrist movements were highly stereotyped across cycles. Visual monitoring (via infrared camera) confirmed the same was true of the arm as a whole (e.g. the lateral position of the elbow was quite stereotyped across revolutions). Eye position and pupil dilation were monitored but are not analyzed here.

### Task

The monitor displayed a virtual landscape, generated by the Unity engine (Unity Technologies, San Francisco). Surface texture and landmarks to each side of a central path provided visual cues regarding movement through the landscape. A salient visual cue (landscape color) indicated whether pedaling must be 'forward' (the hand moved away from the body at the top of the cycle) or 'backward' (the hand moved toward from the body at the top of the cycle) to produce forward progress in the virtual world. Trials were blocked into forward and backward pedaling. Movement was along a linear path. One rotation of the pedal produced one arbitrary unit of movement. A target on the landscape moved continuously 'away' from the monkey's first-person virtual location at a fixed speed, and thus needed to be 'chased'. Reward was delivered every 550ms so long as the monkey's virtual position was close to the target. Recordings were then divided into cycles based on the stereotypy of the x and y positions. The cycles were divided into eight speed bins for each of the two cycling directions, leading to a total of sixteen different conditions.

Artificial viscosity and mass were the same as in *Russo et al., 2018*; *Russo et al., 2020* and modestly supplemented the natural viscosity of the motor and inertia of the apparatus. With no added viscosity and mass the apparatus tended to feel slightly 'slippery' and 'floppy'. We found the device felt more natural to manipulate with the effective viscosity and mass increased modestly. Because added viscosity was modest, it was not the primary limiting factor on top speed. Both when the monkeys performed the task and when we tried it, top speed appeared to be limited (to not much more than 3 Hz) by factors intrinsic to the neuromuscular control of the arm.

### Neural recordings

After initial training, we performed a sterile surgery during which monkeys were implanted with a head restraint. Cylinders (Crist Instruments, Hagerstown, MD) were centered over the border between caudal PMd and primary motor cortex, located according to a previous magnetic resonance imaging scan. Cylinders were placed normal to the cortical surface. The skull within the cylinder was left intact and covered with a thin layer of dental acrylic. Neural recordings were made using conventional single electrodes (Frederick Haer Company, Bowdoinham, ME) driven by a hydraulic microdrive (David Kopf Instruments, Tujunga, CA). Electrodes were introduced through small (3.5 mm diameter) burr holes drilled by hand through the acrylic and skull, under anesthesia. Sequential recording with conventional

electrodes (as opposed to simultaneous recording with an array) allowed us to acquire recordings from a broader range of sites, including sulcal sites inaccessible to many array techniques. Recording locations were guided via microstimulation, light touch, and muscle palpation protocols to confirm the trademark properties of each region. Recordings were made from primary motor cortex (both surface and sulcal) and the adjacent (caudal) aspect of dorsal premotor cortex. For all analyses, these recordings are analyzed together as a single motor cortex population. Motor cortex recordings were restricted to regions where microstimulation elicited responses in shoulder, upper arm, chest and forearm.

Neural signals were amplified, filtered, and manually sorted using a Blackrock Microsystems Digital Hub and 128-channel Neural Signal Processor. A total of 126 isolations were made in monkeys C and D. Nearly all neurons that could be isolated in motor cortex were responsive during cycling. A number of isolations (25) were discarded due to low signal-to-noise ratios or insufficient trial counts. No further selection criteria were applied. For each trial, the spikes of the recorded neuron were filtered with a Gaussian (20ms SD) to produce an estimate of firing rate versus time. These were then averaged across trials as described below.

## EMG recordings

Intra-muscular EMG was recorded from the major muscles of the arm, shoulder, and chest using percutaneous pairs of hook-wire electrodes (30 mm x 27 gauge, Natus Neurology) inserted ~1 cm into the belly of the muscle for the duration of single recording sessions. Electrode voltages were amplified, bandpass filtered (10–500 Hz) and digitized at 1000 Hz. To ensure that recordings were of high quality, signals were visualized on an oscilloscope throughout the duration of the recording session. Recordings were aborted if they contained significant movement artifacts or weak signal. That muscle was then re-recorded later. Offline, EMG records were high-pass filtered at 40 Hz and rectified. Finally, EMG records were smoothed with a Gaussian (20ms standard deviation, same as neural data) and trial averaged (see below). Recordings were made from the following muscles: the three heads of the *deltoid*, the two heads of the *biceps brachii*, the three heads of the *triceps brachii*, *trapezius*, *latissimus dorsi*, *pectoralis*, *brachioradialis*, *extensor carpi ulnaris*, *extensor carpi radialis*, *flexor carpi ulnaris*, *flexor carpi radialis*, and *pronator*. Recordings were made from 1 to 8 muscles at a time, on separate days from neural recordings. We often made multiple recordings for a given muscle, especially those that we have previously noted can display responses that vary with recording location (e.g., the *deltoid*).

## Trial alignment and averaging

The average firing rate was computed across trials with nearly identical behavior. This was achieved by (1) training to a high level of stereotyped behavior, (2) discarding rare aberrant trials, and (3) 'adaptive alignment' of individual trials prior to averaging in each speed bin. First, trials were aligned so that vertical pedal orientation occurred at the same moment for every trial. The trials were then sorted into 1 of 8 total speed bins per cycling direction. Individual trials were then scaled so that all trials in a speed bin had the same duration (set to be the median duration across trials). Trials were not included if scaling changed the time-base by more than 15%. Because monkeys usually cycled at a consistent speed (within a given condition) this brought trials largely into alignment: for example, the top of each cycle occurred at nearly the same time for each trial. The adaptive alignment procedure was used to correct any remaining slight misalignments. The time-base for each trial was scaled so that the position trace on that trial closely matched the average position of all trials. This involved a slight non-uniform stretching, and resulted in the timing of all key moments – such as when the hand passed the top of the cycle – being nearly identical across trials. This ensured that high-frequency temporal response features were not lost to averaging. Any alignment procedure is necessarily imperfect; for example, the non-uniform stretching of time that best aligns position will be slightly different from the non-uniform stretching of time that best aligns velocity. In practice this was a minimal concern. For example, although alignment did not explicitly take into account velocity, velocity was still nicely aligned across trials (*Figure 1b*, top). The same alignment procedure was used for neural and EMG data.

All variables of interest (firing rate, hand position, hand velocity, EMG, etc.) were computed on each trial before adaptive alignment. Thus, the above procedure never alters the magnitude of these

variables, but simply aligns when those values occur across trials. The adaptive procedure was used once to align trials within a condition on a given recording session, and again to align data across recording sessions. This allowed, for example, comparison of neural and muscle responses on a matched time-base.

To calculate the modulation of the firing rate as a function of speed, the dynamic range was first calculated for each neuron and speed by subtracting the minimum firing rate from the maximum firing rate over the course of the trajectory. Next, for each neuron, a line of best fit was fitted to explain how the dynamic range changed as a function of cycling speed. A positive (negative) slope indicated that firing rate increased (decreased) overall with speed.

## Preprocessing and PCA
Because PCA seeks to capture variance, it can be disproportionately influenced by differences in firing rate range (e.g. a neuron with a range of 100 spikes/s has 25 times the variance of a similarly responding neuron with a range of 20 spikes/s). This concern is larger still for EMG, where the scale is arbitrary and can differ greatly between recordings. The response of each neuron / muscle was thus normalized prior to application of PCA. EMG data were fully normalized: $response \leftarrow \frac{response}{range(response)}$, where the range is taken across all recorded times and conditions. Neural data were 'soft' normalized: $response \leftarrow \frac{response}{range(response)+5}$. This is a standard technique (**Russo et al., 2018**) that balances the desire for PCA to explain the responses of all neurons with the desire that weak responses not contribute to the same degree as robust responses. In practice, most neurons had high firing rate ranges during cycling, making soft normalization similar to full normalization.

We used PCA for simplicity, and because it can be readily applied in the same way to neural and muscle recordings, aiding comparison. Related methods such as factor analysis become more appropriate when analyzing single-trial data because they can better model the 'private' spiking variability of each neuron. However, factor analysis provides no advantage over PCA for trial-averaged data that is recorded sequentially. For muscle activity it can be advantageous to use non-negative matrix factorization (**Krouchev et al., 2006**) when activity occurs in relative discrete bursts. Non-negative matrix factorization can ensure that the inferred underlying 'drives' have a similar sensible structure. However, muscle activity was rarely burst-like during cycling; muscles were often modulated fairly continuously throughout the cycle. This makes PCA more natural, especially given the twin desires of capturing maximal variance in as few dimensions as possible, and comparing the resulting trajectories between neural, muscle, and network populations.

## PCA per speed
Neural data (the firing rate of every neuron) for each speed were formatted as a 'full-dimensional' matrix, $X_s^{full}$, of size $n \times T$, where $n$ is the number of neurons and $t \in [1, T]$ indexes across all analyzed times in the trajectory during speed $s$. We similarly formatted muscle data as a matrix, $Z_s^{full}$, of size $m \times T$, where $m$ is the number of muscles. Because PCA operates on mean-centered data, we mean-centered $X_s^{full}$ and $Z_s^{full}$ so that every row had a mean value of zero. PCA was used to find $X_s$, a reduced-dimensional version of $X_s^{full}$ with the property that $X_s^{full} \approx V_s X_s$, where $V_s$ are the speed-specific principal components. PCA was similarly used to find $Z_s$, the reduced-dimensional version of $Z_s^{full}$.

## PCA across conditions
To calculate the 'global subspace', that is the subspace that explains response variance across all speeds, neural data were first formatted as a 'full-dimensional' matrix, $X^{full}$, of size $n \times T$, where $n$ is the number of neurons and $t \in [1, T]$ indexes across all analyzed times and speeds. We mean-centered $X^{full}$ and then used PCA to find $X$, a reduced-dimensional version of $X^{full}$ with the property that $X^{full} \approx VX$, where $V$ are now the 'global' principal components. We employed analogous methods for the muscle data to obtain $Z$, a reduced-dimensional version of $Z^{full}$.

## Recurrent neural networks
We trained recurrent neural networks (RNNs) to produce a target output consisting of the muscle data projected into its top 6 PCs. For most networks we examine, the input was static with a level indicating

the target speed. Networks consisted of 50 units, roughly matching the number of neurons recorded in each monkey. Networks had the following dynamics:

$$r\left(t + \Delta t, c\right) = r\left(t, c\right) + \tfrac{1}{\tau}\left(-r\left(t, c\right) + Af\left(r\left(t, c\right)\right) + Bu\left(t, c\right) + b\right)$$

where $r\left(t, c\right)$ is the network state (the 'firing rate' of every unit) for time $t$ and condition $c$. We used a timestep $\Delta t$ of 4ms, with $\tau = 10$ timesteps. The network was trained on eight conditions, each corresponding to one speed. Thus, $c \in \left[1, 8\right]$. Each condition was presented in multiple 'trials' lasting 2000 timesteps (8 s). The function $f := tanh$ is an element-wise transfer function linking a unit's input to its firing rate. The matrix $A$ captures the connection weights between the network units, and $Bu\left(t, c\right)$ captures the effect of the external input. The vector $b$ captures the bias for each unit. Network output is a linear readout of its firing rates:

$$y\left(t, s\right) = Cf\left(r\left(t, c\right)\right) + d$$

with $d$ representing the bias vector for the output. The parameters $A, B, C, b, d$ were optimized (using TensorFlow's Adam optimizer) to minimize the mean squared error between the network output $y$ and the target output $y_{targ}$ . The external speed-specifying input was of constant amplitude $a$ that was a linear function of the speed: $a = 0.5 + 0.5\frac{c-1}{8}$ . On each trial, the input was always zero for the first 800 timesteps (equivalent to 3.2 s), then 'turned on' (took on the value $a$) to instruct the network to begin to produce an output. After a variable time, the input 'turned off' (became zero), instructing the network to produce no output.

One concern, during training, is that networks may learn overly specific solutions if the number of cycles is small and stereotyped. For example, if only ever asked to produce three 500ms cycles, the network could learn a non-periodic 1500ms trajectory that nevertheless yields a periodic output. Such solutions are degenerate because a network that adopted that solution would be unable to continue producing the periodic output for larger numbers of cycles. To ensure networks could not adopt degenerate solutions, during training we varied the span of time over which the input instructed cycling, from 700 timesteps (2.8 s) to 1100 timesteps (4.4 s). Specifically, the input was of amplitude $a$ from timestep 800 to $n$, where $n$ was sampled from a uniform distribution between 1500 to 1900, that is, $n \sim U\left[1500, 1900\right]$. The input was of amplitude 0 for the rest of the timesteps. The target output was of amplitude 0 when the input was 0. When the input was of amplitude $a$, the output was the muscle activity in the corresponding speed bin $s$, projected into the top 6 PCs and downsampled by a factor of 4 (to match the 4ms timestep). The muscle activity was repeated in time for as many cycles as necessary to fill the time period where the input was 'on' (anywhere from ~3 to~8 cycles). Inspection of both individual-unit responses and the PCs revealed that this training procedure was successful. All networks that learned the task used repeating trajectories, with a period that matched that of the periodic output. There was one exception but it is expected: the first cycle could differ slightly from all the rest, as it took time to settle into the stable limit cycle. This is typical of networks trained on this type of task, and is also observed empirically (*Russo et al., 2018*). For this reason, analyses considered one of the middle cycles.

In the past, we have often found that encouraging noise-robust low tangled solutions requires training in the presence of noise, or the use of some other form of regularization that encourages well-behaved solutions (*Sussillo et al., 2015*). In the present case this was not necessary; the same stacked-elliptical structure emerged in networks trained both in the presence and absence of noise. For networks where we included noise, noise was additive and Gaussian, with variance 0.01 (approximately 2% of the input on each timestep). This was done simply to verify that the same solution occurred in the presence of noise.

## Linear decoding of neural firing rates

Single-neuron firing rates were regressed against the muscle factors and the network factors separately. First, PCA was performed on muscle activity (across speeds and times for the relevant cycling direction) to produce two- or three- dimensional factors. Then, ridge regression was performed using the muscle factors to explain each neuron's firing rate (performed separately for each neuron). The regularization parameter was chosen based on earlier exploration of cross-validation performance. To quantify fit quality, we computed the population $R^2$ : the average $R^2$ across neurons taking into

account different response magnitudes. The above procedure was repeated using the network factors to explain single-neuron responses. For each network, we computed the difference in population $R^2$ when explaining neural activity with network factors versus muscle factors.

## Fit to ellipses

Trajectories were projected into their top 2 PCs. The ellipse that best fit the resulting 2D trajectory was found using least squares (*Gal, 2020*). The $R^2$ of the elliptical fit was calculated by uniformly sampling this ellipse with the same number of time points as the trajectory, then calculating the population $R^2$ across timepoints. Some muscle trajectories did not have an ellipse as a solution, and were instead better fit by a hyperbola (e.g. in cases where the muscle trajectory was saddle-shaped, points were closer to a hyperbola than any ellipse). To be conservative, we discarded these in subsequent tests of the differences in $R^2$ .

## Understanding network structure

To ascertain the presence of a limit cycle, network trajectories were perturbed from their typical trajectory (for that speed input). Perturbations were in a random direction within the top two PCs, described by the random vector $\epsilon$, where $\epsilon \sim U\left[-0.1, 0.1\right]$. This two-dimensional perturbation was transformed into the full-dimensional space (one dimension per neuron) and the network was started from this initial state. The resulting network trajectory across the next 2000 time points was then transformed back into PC space, and it was verified that these returned back to the typical trajectory. This procedure was repeated for different speed inputs.

In a different set of explorations, the network state was perturbed from its typical trajectory in one of two directions. The first (rhythm-generation perturbation) was chosen to maximize impact on the trajectory in the first two PCs while minimizing impact on the muscle readout being examined. The second (muscle-generation perturbation) was chosen to do the opposite: minimize impact on the trajectory in the first two PCs while maximizing impact on that muscle readout.

We tested how networks responded to speed inputs that were not used during training but that were within the range of trained inputs. We tested static speed inputs with values between the eight input levels used during training. We also tested dynamic inputs whose value steadily increased from the smallest to the largest trained input in a continuous ramp.

To train networks that received unique inputs for each speed (*Figure 3—figure supplement 2C, d*), networks were trained with the same equations as above. However, the input $u$ was now a one-hot encoding vector of dimension 8. that is, if the target output was the muscle activity for speed-bin 3, the static input was [0,0,1,0,0,0,0,0].

## Tangling

Trajectory tangling $Q(t)$ is described in detail in *Russo et al., 2018*, and is defined as

$$Q\left(t\right) = \max_{t'} \frac{\|\dot{x}(t) - \dot{x}(t')\|_2^2}{\|x(t) - x(t')\|_2^2 + \epsilon}$$

Here, $x\left(t\right)$ is the neural state at time t, $\dot{x}\left(t\right)$ is its temporal derivative, $\| \cdot \|$ is the Euclidean norm, and $\epsilon$ is a small constant that prevents division by 0. Briefly, the neural state, $x\left(t\right)$ is a vector comprised of the $t^{th}$ column of $X$, where $X$ is the neural activity projected into its principal components. Muscle tangling was computed analogously, based on $Z$. We computed the derivative of the state as $\dot{x}_t = \frac{x_t - x_{t-\Delta t}}{\Delta t}$ , where $\Delta t$ was 1ms. The constant $\epsilon$ (set to 0.1 times the variance of $x$) determines how small the denominator can become, which prevents the denominator from shrinking below 0.1 times the average squared distance from zero. To assess robustness, tangling was typically computed for different dimensionalities of $X$ and $Z$ and in some cases was computed in the full-dimensional space without applying PCA (in which case the neural state is simply a vector of firing rates and the muscle state is simply a vector of muscle-activity values).

## Path similarity

Path similarity was calculated in a two-dimensional subspace. Path similarity quantified the degree to which trajectory shape changed as one moved away from a 'reference speed'. Analysis considered the trajectory for each of the eight speed bins. We refer to the two-dimensional trajectory (in the

top two PCs) for a given speed bin as $X_s$. We picked a particular speed bin to provide the reference trajectory, $X_{s_{ref}}$. $X_{s_{ref}}$ is of size $2 \times T_{s_{ref}}$, where $T_{s_{ref}}$ are the number of milliseconds it took to complete a cycle for that speed bin. Because every speed bin necessarily had a different number of timepoints, once the reference trajectory was chosen, every other $X_s$ was interpolated or compressed (along the time axis) to be of size $2 \times T_{s_{ref}}$. We then quantified the similarity of every $X_s$ to $X_{s_{ref}}$, allowing for a rigid rotation of $X_s$ to minimize the root mean squared (RMS) distance from the reference trajectory $X_{s_{ref}}$. To do so we found the rotation matrix $R$ that minimized the cost function $\min_R \|X_{s_{ref}} - RX_s\|_F$ s.t. $R^T R = I$, $det(R) = 1$. The solution was implemented using singular value decomposition as in the Kabsch algorithm (*Kabsch, 1976*). The coefficient of determination $R^2(X_{s_{ref}}, RX_s)$ between the reference trajectory and the rotated trajectory was calculated. There was one such value of $R^2$ for every possible combination of $X_{s_{ref}}$ and $X_s$. For each $s_{diff} = \{\pm 1, \pm 2, ... \pm 5\}$, we computed the average (and standard error) of the $R^2$ values for all comparisons where $s_{diff} = s - s_{ref}$. For example, for $s_{diff} = 5$, we averaged the $R^2$ values obtained when comparing speed bins 1 vs 6, 2 vs 7, and 3 vs 8. We repeated this analysis, for the neural data, using 2D subspaces defined by higher PCs.

## Speed axis

To create *Figure 6c*, we plotted the data projected into a three-dimensional space spanned by the top two PCs and a 'speed axis'. The speed axis was calculated by finding the one-dimensional axis in the neural data that best decodes the mean angular velocity and is orthogonal to the first two PCs. To restrict analysis to dimensions that captured considerable variance, the neural data was first projected into its top 12 PCs (PCA was performed across all times and speeds). We used the data projected into PCs 3–12 to find the speed axis. We calculated the mean of the neural data in each speed $s$ as $\bar{x}_s = \frac{1}{T_s} \sum_{t=1}^{T_s} x_s(t)$, where $x_s(t)$ is the data for speed $s$ at time $t$, projected into the across-speed PCs 3–12 (and is thus of size 10 × 1). We found the linear decoder $d$ of size 10 × 1 that minimizes the following cost function.

$$\| \begin{bmatrix} \bar{x}_1 & \bar{x}_2 ... & \bar{x}_8 \end{bmatrix} d - \begin{bmatrix} \bar{v}_1 & \bar{v}_2 ... & \bar{v}_8 \end{bmatrix} \|_2$$

$$s.t. \|d\|_2 = 1$$

Here, $\bar{v}_s$ is the mean-centered mean angular velocity of the hand during speed bin $s$. Analogous methods were applied to the muscle data.

## Trajectories with artificially rescaled time

We wanted to test how high trajectory tangling would rise for neural data if trajectories were constrained to follow the same path for all speeds. To simulate such trajectories, we used the path taken during speed bin 6 as the reference path and rescaled time to generate the trajectories at all other speeds. To include the effect of biophysically based noise, we simulated spike trains for each neuron at each speed, using the trajectory as the conditional intensity function. The same number of trials were simulated as the original number of trials for that neuron. The thinning method was used to simulate spike trains; this simulates a point process given a bounded conditional intensity function (*Ogata, 1981*). The same preprocessing steps were applied to these simulated spike trains as the neural recordings, and then trajectory tangling was computed as before. This process ensured that even though the 'true' trajectories were identical across speeds, the 'measured' trajectories were not because they contained some realistic sampling error (without this, trajectory tangling could be inflated relative to real recorded data).

## Separation removed trajectories

Neural trajectories for different speeds differed (typically modestly) in shape and were separated by both a translation (a difference in their mean value) and by occupying different dimensions. To test whether these forms of separation were important for low tangling, we wished to remove them while maintaining trajectory shape. To do so, we used the speed bin 6 as the reference condition and interpolated or compressed the data in other speed bins such that they had the same number of time points as the reference condition. We then applied the optimal rotation to minimize RMS between the trajectories in all speed bins to the trajectory in the reference speed bin using the Kabsch algorithm, in the same way as described in the Path Similarity section of the Methods.

## Distance between trajectories

We chose one speed bin, $s_{ref}$, as the reference condition ($s_{ref} = 4$ for the primary analysis). We interpolated or compressed the data in all other speed bins such that they had the same number of time points as the reference condition. Each time point corresponds to a phase $\theta$ describing the angular displacement of the pedal, with $\theta = 0$ corresponding to the limb being in the vertical position. This allows us to define $x_{s_{ref}}(\theta)$ to be the neural state in the reference trajectory when the pedal is at phase $\theta$. For each other speed bin, we computed the phase-dependent distance between its trajectory and the reference trajectory as $\Delta_{x_s}(\theta) = \|x_{s_{ref}}(\theta) - x_s(\theta)\|_2$. The distance between the means was similarly calculated as $\Delta_{\bar{x}} = \|\bar{x}_{s_{ref}} - \bar{x}_s\|_2$, where $\bar{x}_s = \frac{1}{T_s}\sum_{t=1}^{T_s} x_s(t)$.

## Additional information

### Funding

| Funder | Grant reference number | Author |
|---|---|---|
| Grossman Center for the Statistics of Mind | | Mark M Churchland |
| Alfred P. Sloan Foundation | FG-2015-65496 | Mark M Churchland |
| Simons Foundation | 542963 | Mark M Churchland |
| NIH | 1U19NS104649 | Mark M Churchland |
| NIH | 5T32NS064929 | Abigail A Russo |
| Kavli Foundation | | Mark M Churchland |
| Simons Foundation | 325171 | Mark M Churchland John Cunningham |
| Swiss National Science Foundation | P2SKP2 178197 | Shreya Saxena |
| Swiss National Science Foundation | P400P2 186759 | Shreya Saxena |

The funders had no role in study design, data collection and interpretation, or the decision to submit the work for publication.

### Author contributions

Shreya Saxena, Conceptualization, Formal analysis, Investigation, Writing – original draft, Writing – review and editing; Abigail A Russo, Conceptualization, Data curation, Formal analysis, Investigation, Methodology, Writing – original draft; John Cunningham, Conceptualization, Supervision, Writing – original draft; Mark M Churchland, Conceptualization, Funding acquisition, Investigation, Methodology, Project administration, Resources, Supervision, Visualization, Writing – original draft, Writing – review and editing

### Author ORCIDs

Shreya Saxena  http://orcid.org/0000-0003-4655-7050
Mark M Churchland  http://orcid.org/0000-0001-9123-6526

### Ethics

All protocols were in accord with the National Institutes of Health guidelines and approved by the Columbia University Institutional Animal Care and Use Committee. (Protocol number AC-AABE3550).

### Decision letter and Author response

Decision letter https://doi.org/10.7554/eLife.67620.sa1
Author response https://doi.org/10.7554/eLife.67620.sa2

## Additional files

### Supplementary files
• Transparent reporting form

### Data availability
Neural and EMG data have been deposited in figshare: https://figshare.com/s/b2a0557c239a1010d8ea.

The following datasets were generated:

| Author(s) | Year | Dataset title | Dataset URL | Database and Identifier |
|-----------|------|---------------|-------------|-------------------------|
| Russo AA, Churchland MM | 2022 | Cousteau_cn_new.mat | https://figshare.com/s/b2a0557c239a1010d8ea?file=34512257 | figshare, 34512257 |
| Russo AA, Churchland MM | 2022 | Drake_cn_new.mat | https://figshare.com/s/b2a0557c239a1010d8ea?file=34512260 | figshare, 34512260 |

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
