## [Editor Report]

This elegant study furthers our understanding about the mechanisms by which distributed systems control rhythmic movements of different speeds. The authors trained an artificial recurrent neural network to produce muscle activity patterns similar to those that monkeys generate when performing an arm cycling task at different speeds. The dominant patterns in the neural network do not directly reflect muscle activity, and these dominant patterns do a better job than muscle activity at capturing key features of neural activity recorded from the monkey motor cortex in the same task. In addition to the main result, the study provides a particularly clear example of how thinking in terms of network dynamics can naturally explain empirical observations in terms of the computation being performed.

---

## [Decision Letter]

**Decision letter after peer review:**

Thank you for submitting your article "Motor cortex activity across movement speeds is predicted by network-level strategies for generating muscle activity" for consideration by *eLife*. Your article has been reviewed by 3 peer reviewers, and the evaluation has been overseen by Andrew Pruszynski as Reviewing Editor and Joshua Gold as Senior Editor. The reviewers have opted to remain anonymous.

Essential revisions:

1) The reviewers raise a number of important points about the RNN, the characterization of its dynamics, and its relationship to the empirical data. This includes further details about the network and analysis but also a broader perspective on the interpretation. Three notable considerations along these lines are: (a) establishing whether strong interactions between the "dominant" dimensions generate the rhythm while muscle-related dimensions do not push on these dominant dimensions; (b) further consideration about the constant inputs into the network and the implications of this choice in terms of the observed relationship to M1 activity and overall interpretation; (c) explicitly laying out what muscle like commands would look like.

2) The long-term impact of this study could be increased by providing a more comprehensive high-level discussion about the relationship between the extracted PCs and the computations done in motor cortex. Use this as an opportunity to link the present work to more traditional concepts in the field like the representation of motor commands.

*Reviewer #1 (Recommendations for the authors):*

1. Was the torque applied by the animals to the handle measured at each speed and is an estimate of the handle's damping ratio available?

2. The authors state that data windows of variable length were used "to avoid overly specific solutions." Was this done to enforce a return to a stable fixed point when the inputs switch to zero, independently of the movement phase at which this switch occurred?

3. The authors may wish to comment on whether the RNN dynamics in the dominant dimensions might be topologically equivalent to a simple canonical model. Intuitively, it looks like the network trajectories might obey something like θ' = cs, r' = r(1 – r^2), z' = s – z, where θ is the angle in the x-y plane, r is the radius, z is the vertical dimension, s is the speed input, and c is a constant. Establishing the existence of smooth transformations from the vector fields learned by the networks to a canonical model would bolster the claim that the networks have found the same general solution, but this analysis is not critical.

4. More details on the RNN and fitting could be provided in the Methods. What values were chosen for the time constant τ and time step Δt? If I understand correctly, the node state does not depend explicitly on speed, so the notation v(t,s) (line 754) was slightly confusing.

*Reviewer #2 (Recommendations for the authors):*

How do the stacked ellipses observed here resemble or not resemble the spiral trajectories observed in SMA during motor sequences?

*Reviewer #3 (Recommendations for the authors):*

This study is a further examination of the dynamical-systems (non-representational) view of the motor cortex, and its relation to motor execution and muscle activity. It combines modeling with single-electrode neural and EMG recordings in two monkeys, using the cycling paradigm that this group has developed over the last number of years. In particular, they examine the question how each of these signals (including modeled manifold solutions) change with the speed and direction of pedaling. The experiments are well designed and the paper is clearly written, with the occasional lapse into excessive neural or mathematical jargon. In addition to the central questions the authors pose, it provides a good discussion of the nature of these dynamical models, what we can learn from them, and what is still unknown. I have no major concerns.

The one methodical approach that should be presented more clearly is that of the "population recordings" from M1:

22: "…we recorded motor cortex population activity during the same task."

76: "We compared network solutions with empirical population activity recorded from motor cortex.

Describing post-hoc time-aligned single-electrode recordings this way is a bit of a stretch. I was actually confused for a while, assuming initially that the neural trajectories must all have been from the ANNs. It is compounded by the fact that there is no explicit discussion of this issue, which I would recommend adding. The same concern applies to the EMGs, though less so. I doubt there are any significant issues, but it should be acknowledged.

In a similar vein, I was occasionally lost, trying to figure out which results came from the artificial networks and which from electrodes. There is at least one reference to "empirical" networks, an adjective I've never found terribly useful as a reference to something in the brain as opposed to the computer (perhaps, "recorded", "actual"?). This was mostly an issue when I was thinking the manifold responses must all from the networks, with comparisons make to only single electrodes, but it might help to make more explicit reference consistently to ANNs.

I find the relation between the initial (elliptical, non-muscle-like) PCs and the higher-order ones intriguing. Are the former "pulling along" the latter to produce a muscle-like output? Are they somehow doing separate computations? is it all just a computational trick that separates them?

One reference to the literature that gets swept aside to quickly is this:

485: "For example, autonomous dynamics provide a poorer fit to the data during grasping…"

To the extent that this comment reflects the effect of afferent input during object manipulation ("Tangling will also be high when unpredictable external inputs dominate…"), it should be noted that their data did not include contact. This observation remains an important one.

---

## [Author Response]

Essential revisions:1) The reviewers raise a number of important points about the RNN, the characterization of its dynamics, and its relationship to the empirical data. This includes further details about the network and analysis but also a broader perspective on the interpretation. Three notable considerations along these lines are:(a) establishing whether strong interactions between the "dominant" dimensions generate the rhythm while muscle-related dimensions do not push on these dominant dimensions;

We agree this is important to explore and have now added a new analysis to do so (Figure 3f). The results are very pretty. Things do indeed work as the reviewers state. Pushing the network in the dominant dimensions – i.e., along the elliptical limit cycle – primarily causes a change in the phase of the rhythm. In contrast, pushing the network in a muscle-related dimension causes an immediate large change in muscle activity that quickly dissipates, with no subsequent change in the phase of the rhythm. This is now explained in the text:

“The ability of elliptical network trajectories to generate non-elliptical muscle trajectories seems counter-intuitive but is expected (Russo et al. 2018). Elliptical trajectories set the basic rhythm and provide the fundamental frequency, while the ‘muscle-encoding’ signals that support complex outputs ‘live’ in dimensions beyond the first three PCs. To illustrate, we consider a single cycling speed (Figure 3f) and perturb network activity either along the elliptical path but orthogonal to the muscle readout (*orange*) or in a direction that overlaps with the muscle readouts but is orthogonal to the elliptical path (*green*; note that this direction involves PCs beyond the first three and is not the same direction that separates trajectories across speeds). As expected, perturbations that overlapped muscle readouts caused a large immediate change in muscle readout (*green trace*). Muscle readouts then rapidly returned to normal, after which the rhythm continued at its original phase. Thus, these dimensions are critical for network outputs, but interact little with the dynamics that set the overall rhythm. In contrast, perturbing along the elliptical trajectory created a much smaller immediate effect, but permanently altered output phase (*orange trace*). Thus, the network solution involves two components. The phase of the elliptical trajectory sets the phase of the output. Muscle readouts draw from the elliptical trajectory, but also draw heavily from orthogonal dimensions that contain smaller higher-frequency signals. It is these smaller off-ellipse features that allow the network to generate non-sinusoidal activity patterns that are different for each muscle. This is a natural strategy that allows a simple, stable, elliptical trajectory to generate multiple temporally structured outputs.”

(b) further consideration about the constant inputs into the network and the implications of this choice in terms of the observed relationship to M1 activity and overall interpretation;

We agree this is important. There are two classes of non-constant inputs to consider: (1) sensory feedback and (2) non-constant input commands from ‘higher’ motor areas. We have added simulations to explore both.

Regarding the first input type, M1 presumably receives continuous proprioceptive feedback throughout the cycle. It is thus critical to know that the stacked-elliptical solution continues to be relevant in the presence of such feedback. (Certainly it would complicate interpretation if a solution incompatible with feedback were present in a brain area that receives feedback).

We thus trained networks in the presence of simulated sensory feedback. Because the exact form of sensory feedback is unknown, we simulated a few potential forms. Networks continued to use the same stacked-elliptical solution in the presence of feedback. This ceased to be true only if feedback was so strong that it dominated activity. These results are intuitive and agree with prior observations. Modest levels of feedback simply become one more form of recurrence that contributes to the network’s dynamics, consistent with Sussillo et al. 2015. However, if feedback is overly strong, network activity effectively becomes a representation of sensory variables and solutions are no longer realistic. This agrees with the presence of high tangling in sensory areas including S1 (Russo et al. 2018). The new analysis is in Figure 9, and is discussed in the new section *Generality of the network solution*, and again in the Discussion.

Regarding the second input type, the incoming motor commands that ‘tell’ M1 what to do may not be static. For example, it is quite plausible that M1 receives a simple rhythm from SMA, to which it phase-locks its output. We thus simulated networks that received a pair of sinusoidal inputs (in phase quadrature) that implicitly specified speed via their frequency. Interpretation of these results is a bit complicated: networks did not consistently produce the empirical stacked-elliptical structure, but they often did. What can be logically concluded is the following. (1) The empirical data is certainly consistent with networks receiving a simple rhythm, because such networks often use the stacked-elliptical solution. (2) Given the data, one *probably* leans towards the idea of a static input, because such networks always produced the empirically realistic solution. These new simulations are discussed at the end of the section ‘*Understanding network solutions*’.

As an aside, there exist other possible inputs that could be worth exploring. For example, we suspect that realistic solutions would consistently emerge in networks that receive *both* a static speed-specifying command (as in our ‘main’ network simulations) and rhythmic commands (as discussed above). However, we decided we wished to avoid being pulled down this rabbit hole. Part of our point is that the empirical stacked-elliptical solution is expected under a broad range of reasonable assumptions. The upside of this generality is that it allows the network perspective to make clear predictions regarding the empirical data. The downside is that, when those predictions are confirmed, that confirmation supports the general hypothesis but doesn’t tightly constrain the exact modeling choices. There are a few choices that clearly *won’t* work (e.g., speed-specific inputs) but also many choices that work well. Further experiments (e.g., multi-area recordings) would be needed if one were to try and determine specifics regarding inputs.

(c) explicitly laying out what muscle like commands would look like.

We have removed the term ‘muscle-like’ from the manuscript. We had originally used this term to try and balance two things: (1) we think it likely that outgoing cortical commands effectively constitute commands for muscle activity, and also (2) this can’t be precisely true, because there will inevitably be transformations in the spinal cord. Because the nature and extent of that transformation is unknown, we had used the term ‘muscle-like’ as a hedge. Networks are trained to produce the actual empirical muscle activity, but we assume that this is simply a reasonable approximation. We used the term ‘muscle-like’ to try and convey this, but we agree it was a bit confusing. It thus seemed best to simply remove this term and say what we mean. For example, in the new section ‘*Generality of the network solution*’ we avoid the term muscle-like and instead say:

“Meaningful transformations presumably occur between cortical outputs and motoneuron activity (Shalit et al. 2012; Albert et al. 2020), and motoneurons receive other sources of drive. One thus wishes to know that network predictions are relatively insensitive to assumptions regarding the exact output, and this was indeed the case. This insensitivity relates to the form of the solution. The dominant elliptical trajectory ensures low tangling and can provide the fundamental frequency of the output. All other aspects of the output are built from smaller signals that lie in dimensions orthogonal to the dominant trajectory.”

2) The long-term impact of this study could be increased by providing a more comprehensive high-level discussion about the relationship between the extracted PCs and the computations done in motor cortex. Use this as an opportunity to link the present work to more traditional concepts in the field like the representation of motor commands.

There are a few possible interpretations of this request. We are a little unsure which was meant, but we like them all so we have made multiple changes.

First, the level of PCs can often seem abstract, in contrast to the seeming simplicity of representational explanations. This is a historical / sociological truth rather than an epistemic truth. But it is still true. We have thus added a paragraph to ground our motivation for the PCA based approach. The new paragraph is at the beginning of the section ‘*Understanding network solutions*’:

“Recurrent-network solutions tend to have the following useful characteristic: a basic understanding doesn’t require considering every unit and connection, but can be obtained by considering a smaller number of factors, each a weighted sum of the activity of all units (Sussillo and Barak 2013; DePasquale, Churchland, and Abbott 2016; Maheswaranathan et al. 2019; Mante et al. 2013). By ascertaining how and why network-state trajectories behave in this ‘factor space’, one can often determine how the network solves the task. If factors are defined wisely, the response of each individual unit is approximately a weighted sum of factors. Thus, if one understands the factors, individual-unit responses are no longer mysterious. There are many reasonable ways of obtaining factors, but PCA is commonly used because it ensures factors will be explanatory of single-unit responses (‘maximizing captured variance’ is equivalent to ‘minimizing single-neuron reconstruction error’). We used PCA above to identify the network factors, and show that they are explanatory not only of their own single-unit responses, but of empirical single-neuron responses as well. In contrast, purely representational muscle-based factors were less successful. However, knowing that network factors are explanatory means little if one doesn’t understand why those factors behave the way they do. Thus, we now turn to the task of understanding network solutions in the factor-space obtained by PCA.”

Second, while most aspects of network activity are not representational, networks definitely do still contain representations. After all, the network contains a representation of muscle activity that can be linearly read out! It is just that the representational signals are typically small in most networks; the top PCs are typically dominated by the need for well-behaved dynamics, and the smaller PCs do the ‘encoding’. We have made multiple changes to the manuscript to highlight this, including adding the following section:

“In our networks, each muscle has a corresponding network dimension where activity closely matches that muscle’s activity. These small output-encoding signals are ‘representational’ in the sense that they have a consistent relationship with a concrete decodable quantity. In contrast, the dominant stacked-elliptical structure exists to ensure a low-tangled scaffold and has no straightforward representational interpretation.”

Third, there exist other important traditional concepts in our field such as the idea of feedback control. (In our view feedback-control frameworks have been more explanatory, and certainly make far more sense, than purely representational frameworks). It is thus reasonable to want to link to feedback control ideas, and in particular to the modern conception that corrections are flexibly tailored to the needs of the present movement. We have thus added a paragraph on this topic:

“We did not attempt to simulate feedback control that takes into account unpredictable sensory inputs and produces appropriate corrections (Stavisky et al. 2017; Pruszynski and Scott 2012; Pruszynski et al. 2011; Pruszynski, Omrani, and Scott 2014). However, there is no conflict between the need for such control and the general form of the solution observed in both networks and cortex. Consider an arbitrary feedback control policy:z=gc(t,uf) where uf is time-varying sensory input arriving in cortex and is a vector of outgoing commands. The networks we trained all embody special cases of the control policy where uf is either zero (most simulations) or predictable (Figure 9) and the particulars of *z* vary with monkey and cycling direction. The stacked-elliptical structure was appropriate in all these cases. Stacked-elliptical structure would likely continue to be an appropriate scaffolding for control policies with greater realism, although this remains to be explored.”

In ongoing work we have simulated networks that instantiate flexible feedback control. So we know this class of networks can do so. However, we do not include any of those simulations because they are beyond the scope of the present study.

Reviewer #1 (Recommendations for the authors):1. Was the torque applied by the animals to the handle measured at each speed and is an estimate of the handle's damping ratio available?

We did initially use a torque sensor inline with the motor. However, we found torque records to be unhelpful in assessing what one really cares about: the force produced by the monkey’s own muscles. In retrospect this is kind of obvious. For example, on the way ‘back up’ to the top of the cycle, measured torque greatly underestimates muscle force, much of which is being ‘spent’ lifting the arm against gravity. The opposite happens on the way down. Our solution to this was simple: record the muscle activity itself.

Regarding viscosity, we assume the reviewer wishes to know because, if viscosity were high, that would make cycling very difficult at high speeds. This is a good point, and we should have clarified that the added viscosity was quite modest. We didn’t add viscosity to increase effort. Indeed, top speed is limited by neuromuscular control within the arm itself, rather than by the apparatus. We just added a little viscosity to make the pedal feel natural. One wants to feel a bit of resistance, else it feels like cycling on an ice rink. It is hard to explain without trying it yourself but it really does feel better with just a bit of added viscosity. Importantly, this was modest and wasn’t what limited the ability to cycle quickly. To clarify, we have added the following to the Methods:

“Artificial viscosity and mass were the same as in CITE RUSSO ET AL. 2018 and 2020 and modestly supplemented the natural viscosity of the motor and inertia of the apparatus. With no added viscosity and mass the apparatus tended to feel slightly ‘slippery’ and ‘floppy; we found the device felt more natural to manipulate with these added forces. Added viscosity was modest and thus not the primary limiting factor on top speed, which appeared (both in the monkeys and when we tried it) to be limited to not much more than 3 Hz by factors intrinsic to the neuromuscular control of the arm.”

The viscosity was the same as in Russo et al. 2018 and 2020. It was expressed in terms of volts (sent to the motor) per cycle/s, so it is hard to provide a number in terms of torque per degree/s. We also don’t know the internal viscosity of the motor and chain system. In principle one could attempt to measure this physically, but this would be involved. And of course the arm itself effectively has its own viscosity. For these reasons we just punted, and decided to always measure muscle activity.

2. The authors state that data windows of variable length were used "to avoid overly specific solutions." Was this done to enforce a return to a stable fixed point when the inputs switch to zero, independently of the movement phase at which this switch occurred?

A good question, and closely related to comment 5 above. This is now clarified in the Results:

“Networks were trained across many trials, each of which had an unpredictable number of cycles. This ensured networks could not learn a non-periodic solution, which would be possible if the number of cycles were fixed and small.”

An improved unpacking of this reasoning is provided in the Methods.

3. The authors may wish to comment on whether the RNN dynamics in the dominant dimensions might be topologically equivalent to a simple canonical model. Intuitively, it looks like the network trajectories might obey something like θ' = cs, r' = r(1 – r^2), z' = s – z, where θ is the angle in the x-y plane, r is the radius, z is the vertical dimension, s is the speed input, and c is a constant. Establishing the existence of smooth transformations from the vector fields learned by the networks to a canonical model would bolster the claim that the networks have found the same general solution, but this analysis is not critical.

Yes, a simply canonical model like that is roughly right, although one would have to add another dimension or two into which the modest ‘tilt’ across speeds can happen. (We think this still matters topologically, although to be fair one of us is consistently confused by topology).

4. More details on the RNN and fitting could be provided in the Methods. What values were chosen for the time constant τ and time step Δt? If I understand correctly, the node state does not depend explicitly on speed, so the notation v(t,s) (line 754) was slightly confusing.

We agree this section was not well written and we apologize for not specifying tau or the timestep (the timestep was 4 ms, and the tau was ten timesteps and thus 40 ms). This section has been rewritten. The use of ‘s’ as an index has been changed to ‘c’. The use of ‘c’ denotes ‘condition’ (which is stated explicitly) which is a more natural index. (We agree that ‘s’, while formally correct, was indeed confusing).

Reviewer #2 (Recommendations for the authors):How do the stacked ellipses observed here resemble or not resemble the spiral trajectories observed in SMA during motor sequences?

A great (and subtle) question. We have added the following to the Discussion:

This reinforces a point made above: the dominant features of network solutions often depend on the nature of a computational problem, rather than the specific input-output function or the particular domain (e.g., motor versus cognitive). Similar trajectory geometries can even serve very different computational purposes. For example, in the supplementary motor area, individual cycles are separated as a function of progress within a larger action (e.g., cycling exactly seven cycles and then stopping), producing a helical trajectory across time. In motor cortex, our present model predicts helical structure with steadily increasing speed (something we have informally observed empirically during different experiments). These helical solutions have something high-level in common, yet the nature of the underlying computation is of course completely different (keeping track of progress, versus specifying speed). This underscores that certain trajectory ‘motifs’ are broadly useful, and constrain but don’t fully specify the underlying computation.

Reviewer #3 (Recommendations for the authors):This study is a further examination of the dynamical-systems (non-representational) view of the motor cortex, and its relation to motor execution and muscle activity. It combines modeling with single-electrode neural and EMG recordings in two monkeys, using the cycling paradigm that this group has developed over the last number of years. In particular, they examine the question how each of these signals (including modeled manifold solutions) change with the speed and direction of pedaling. The experiments are well designed and the paper is clearly written, with the occasional lapse into excessive neural or mathematical jargon. In addition to the central questions the authors pose, it provides a good discussion of the nature of these dynamical models, what we can learn from them, and what is still unknown. I have no major concerns.

We thank the reviewer for the supportive comments and nice summary.

The one methodical approach that should be presented more clearly is that of the "population recordings" from M1:22: "…we recorded motor cortex population activity during the same task."76: "We compared network solutions with empirical population activity recorded from motor cortex.Describing post-hoc time-aligned single-electrode recordings this way is a bit of a stretch. I was actually confused for a while, assuming initially that the neural trajectories must all have been from the ANNs. It is compounded by the fact that there is no explicit discussion of this issue, which I would recommend adding. The same concern applies to the EMGs, though less so. I doubt there are any significant issues, but it should be acknowledged.

We agree that the term ‘population recordings’ evokes the idea of simultaneity and thus should be avoided when describing sequentially recorded populations. For that reason, we have edited both those sentences to remove the term “population”. We don’t think we ever used the term ‘population recordings’ but if there are any errant occurrences please note them and we will correct them. We agree that is not quite right. At the same time, it is certainly fair to talk about population analyses, so long as the method for constructing the population response is clearly laid out. We apologize that this was not clear. We have added the following to the section of the Results where population analyses are first applied:

“Neural population activity, for a given time and speed, was defined as the trial-averaged response of every recorded neuron for the relevant time within the relevant speed-bin. Muscle population activity was defined analogously. The consistency of behavior (Figure 1b) made it reasonable to combine trial-averaged responses from sequentially recorded neurons (and muscles) into a unified population response.”

Earlier in the Results we stated “Well-isolated single units… were sequentially recorded from motor cortex.” The new text clarifies how we get from single-neuron to population analyses. It is the identical procedure we would have used for simultaneous recordings.

As an aside, when responses are trial-averaged responses and behavior is consistent, we do not believe that one approach (simultaneous vs sequential) is necessarily better. Both are good ways of constructing a population response that can be fruitfully analyzed. Concerns about behavioral drift are reduced with simultaneous recordings, but isolation quality and curation tends to be better with sequential recordings (arguably this is less true with the advent of neuropixels, but it is very true w.r.t. implanted arrays). The key reason for calling something a population analysis, both in our opinion and given historical usage, is that one considers the joint response of many neurons to an identical set of stimuli. For example, this was the way the term was used in Churchland et al. 2001 (J. Neurosci) and w.r.t. the ‘population vector’ of Georgopoulos.

In a similar vein, I was occasionally lost, trying to figure out which results came from the artificial networks and which from electrodes. There is at least one reference to "empirical" networks, an adjective I've never found terribly useful as a reference to something in the brain as opposed to the computer (perhaps, "recorded", "actual"?). This was mostly an issue when I was thinking the manifold responses must all from the networks, with comparisons make to only single electrodes, but it might help to make more explicit reference consistently to ANNs.

We apologize. We have changed that sentence to read:

“Goal-driven networks -- i.e., networks trained to perform a computation intended to be analogous to that performed by a biological neural population”.

We have adjusted the text throughout to avoid creating confusion. The text added in response to the comment above is already a large help, because it clarifies, at the key moment, what we are analyzing.

I find the relation between the initial (elliptical, non-muscle-like) PCs and the higher-order ones intriguing. Are the former "pulling along" the latter to produce a muscle-like output? Are they somehow doing separate computations? is it all just a computational trick that separates them?

These are great questions. We have added an analysis (of the network) to clarify this. This is in Figure 3f. (Also see response to Essential revisions 1a).

We fully endorse the idea that the relationship between the dominant and higher-order PCs is intriguing! Much of our present work involves attempting to more fully understand that relationship.

One reference to the literature that gets swept aside to quickly is this:485: "For example, autonomous dynamics provide a poorer fit to the data during grasping…"To the extent that this comment reflects the effect of afferent input during object manipulation ("Tangling will also be high when unpredictable external inputs dominate…"), it should be noted that their data did not include contact. This observation remains an important one.

A good point. Our wording didn’t properly convey the important difference between tangling that results from unpredictable sensory inputs and tangling that arises from incoming external ‘commands’. The relevant section has been revised and expanded.

“Yet even in a network that relies on strong dynamics, low trajectory tangling is neither always desirable nor always expected. There can be moments where it is necessary for activity to be dominated by ‘unexpected’ inputs… Similar considerations may explain why trajectory tangling is low (and dynamical fits good) in motor cortex during reaching but not during grasping (Suresh et al. 2019). As suggested in that study, grasping may require a more continuous flow of guiding inputs from the rest of the brain.”